# Early maternal loss leads to short- but not long-term effects on diurnal cortisol slopes in wild chimpanzees

**Cédric Girard-Buttoz[1,2]\*, Patrick J Tkaczynski[1,2], Liran Samuni[2,3,4], Pawel Fedurek[5], Cristina Gomes[6], Therese Löhrich[7,8], Virgile Manin[1,2], Anna Preis[3], Prince F Valé[2,3,9,10], Tobias Deschner[11], Roman M Wittig[1,2], Catherine Crockford[1,2,12]**

[1]Department of Human Behavior, Ecology and Culture, Max Planck Institute for Evolutionary Anthropology, Leipzig, Germany; [2]Taï Chimpanzee Project, Centre Suisse de Recherches Scientifiques, Abidjan, Côte d'Ivoire; [3]Department of Primatology, Max Planck Institute for Evolutionary Anthropology, Leipzig, Germany; [4]Department of Human Evolutionary Biology, Harvard University, Cambridge, United States; [5]Division of Psychology, University of Stirling, Stirling, United Kingdom; [6]Tropical Conservation Institute, Florida International University, Miami, United States; [7]World Wide Fund for Nature, Dzanga Sangha Protected Areas, Bangui, Central African Republic; [8]Robert Koch Institute, Epidemiology of Highly Pathogenic Microorganisms, Berlin, Germany; [9]Centre Suisse de Recherches Scientifiques en Côte d'Ivoire, Abidjan, Côte d'Ivoire; [10]Unité de Formation et de Recherche Biosciences, Université Félix Houphouët Boigny, Abidjan, Côte d'Ivoire; [11]Interim Group Primatology, Max Planck Institute for Evolutionary Anthropology, Leipzig, Germany; [12]Institut des Sciences Cognitives, CNRS, Lyon, France

**\*For correspondence:**
cedric_girard@eva.mpg.de

**Competing interest:** The authors declare that no competing interests exist.

**Abstract** The biological embedding model (BEM) suggests that fitness costs of maternal loss arise when early-life experience embeds long-term alterations to hypothalamic-pituitary-adrenal (HPA) axis activity. Alternatively, the adaptive calibration model (ACM) regards physiological changes during ontogeny as short-term adaptations. Both models have been tested in humans but rarely in wild, long-lived animals. We assessed whether, as in humans, maternal loss had short- and long-term impacts on orphan wild chimpanzee urinary cortisol levels and diurnal urinary cortisol slopes, both indicative of HPA axis functioning. Immature chimpanzees recently orphaned and/or orphaned early in life had diurnal cortisol slopes reflecting heightened activation of the HPA axis. However, these effects appeared short-term, with no consistent differences between orphan and non-orphan cortisol profiles in mature males, suggesting stronger support for the ACM than the BEM in wild chimpanzees. Compensatory mechanisms, such as adoption, may buffer against certain physiological effects of maternal loss in this species.

## Introduction

In mammals, mothers are essential for the early development of their infants since they provide postnatal care (*Maestripieri and Mateo, 2009*). Maternal loss in mammals reduces growth (*Samuni et al., 2020*), survival (*Watts et al., 2009*; *Andres et al., 2013*; *Tung et al., 2016*; *Stanton et al., 2020*), and long-term reproductive success (*Andres et al., 2013*; *Strauss et al., 2020*; *Crockford et al., 2020*; *Zipple et al., 2021*, reviewed in *Clutton-Brock, 2016*).

The biological embedding model (BEM; *Power and Hertzman, 1997*; *Miller et al., 2011*; *Berens et al., 2017*) posits that adversity experienced early in life, including exposure to severe stressors, can have deleterious consequences on an individual's physiology and health across their lifespan. The BEM provides a promising conceptual framework to investigate the mechanisms underlying the fitness costs of maternal loss or other forms of early-life adversity.

Early-life adversity impacts several interconnected physiological pathways (reviewed in *Berens et al., 2017*) among which the hypothalamic-pituitary-adrenal (HPA) axis plays a central role (*Miller et al., 2009*; *Miller et al., 2011*; *Taylor et al., 2011*). The HPA axis is activated in response to internal physiological challenges and external stressors through a chain of reactions, known as the 'stress response' or 'reactive homeostasis' (*Romero et al., 2009*), which also results in the release of glucocorticoids (*Sapolsky, 2002*). Overall, this response is adaptive (*Charmandari et al., 2005*): it stimulates the release of energy out of storage in the form of glucose, and it increases cardiovascular circulation and respiratory rate, allowing organisms to respond to acute stressors such as predators (*Sapolsky, 2002*). Exposure to harsh social conditions during childhood, such as those that can result from maternal loss, may lead to repeated and prolonged activation of the HPA axis early in life. These activations provide an adaptive physiological response by mobilizing energy that helps children to cope with the immediate socioecological challenges but may result in long-term HPA axis dysfunction (i.e., either hypo- or hyper-responsiveness to stressor; *Miller et al., 2011*; *Ehrlich et al., 2016*; *Berens et al., 2017*). The HPA axis is considered to be at the core of the link between early-life adversity and fitness since repeated activation of the HPA axis over prolonged periods (chronic stress) and/or HPA axis malfunctioning can have detrimental effects on individual overall health (*Sapolsky, 2002*; *Slavich and Cole, 2013*). For instance, over- and/or prolonged activation of the HPA axis is known to suppress the immune system (*Grossman, 1985*; *Setchell et al., 2010*; *Slavich and Cole, 2013*) and reduce survival (*Campos et al., 2021*). The HPA axis also mediates some of the observed negative effects of early-life adversity on the immune response, such as elevated levels of inflammatory markers in the blood (*Danese et al., 2011*; *Ehrlich et al., 2016*; *Rasmussen et al., 2019*, reviewed in *Berens et al., 2017*). Assessing the consequences of traumatic early-life events, such as maternal loss, on the functioning of the HPA axis can provide insight into the mechanisms underlying the documented fitness costs of such events.

An alternative framework, the adaptive calibration model (ACM), proposes that intra- and inter-individual changes in stress responsivity are mainly always adaptive, allowing individuals to adjust their physiology during development to respond to changes in the social and ecological environment (*Del Giudice et al., 2011*). According to the ACM, modification of the HPA axis activity in response to maternal loss should be sensitive to the phase of development at which maternal loss occurred, especially in long-lived species with an extended immature period. These modifications should then also be more or less long-lasting depending on the social and ecological environment faced by the developing individual after maternal loss (e.g., the amount of support provided by conspecifics).

In humans, maternal loss leads to short- and long-term alterations of the HPA axis functioning (*Heim and Nemeroff, 2001*; *Sánchez et al., 2001*). These effects are typically studied by investigating patterns of cortisol secretion, the main glucocorticoid circulating in mammals, including humans. In humans, cortisol levels follow a diurnal pattern characterized by an early morning peak (awakening response) and a regular decline throughout the day (*Doman et al., 1986*). These diurnal cortisol slopes, as well as the cortisol-awakening response specifically, serve as health markers. Deviations from stereotypical patterns of high morning and low evening cortisol levels (i.e., flatter diurnal slopes) are typically interpreted as indications of pathology and HPA axis dysregulation and/or a marker of chronic stress in human clinical studies (*Pruessner et al., 1999*; *Sánchez et al., 2001*; *Clow et al., 2004*; *Kudielka et al., 2006*; *Miller et al., 2007*). Flattening of diurnal cortisol slopes reflects a compression in the dynamic range of the HPA axis functioning (*Karlamangla et al., 2019*) that is indicative of a reduced ability to respond appropriately to stressors and to downregulate hormonal stress levels. Flatter diurnal cortisol slopes are even associated with direct fitness costs in humans such as reduced survival (*Sephton et al., 2000*).

To test conclusively the BEM and the ACM, which are diverging in some predictions (see above) but are not mutually exclusive, the physiological consequences of maternal loss must be investigated both during childhood and adulthood. Studies during childhood and in the time directly following maternal loss allow the assessment of the proximate responses of orphans in coping with new social challenges,

while studies during adulthood allow the assessment of the long-lasting effects of early-life adversity. In humans, orphaned children typically exhibit lower cortisol-awakening responses (*Carlson and Earls, 1997*) or higher evening cortisol levels (*Gunnar et al., 2001*) than mother-reared children, leading to overall flatter diurnal cortisol slopes (*Carlson and Earls, 1997*; *Tarullo and Gunnar, 2006*). Flatter diurnal slopes and/or lower morning cortisol levels are also observed in children experiencing other forms of early-life adversity, such as maltreatment by parents, parental divorce, or placement in foster families (*Kaufman, 1991*; *Hart et al., 1996*; *Meinlschmidt and Heim, 2005*; *Dozier et al., 2006*; *Bernard et al., 2015*; *McLachlan et al., 2016*). Beyond diurnal cortisol patterns, maternal loss can also be associated with generally higher cortisol levels throughout the day (*Gunnar et al., 2001*).

In support of the BEM, several studies in humans document long-lasting effects of early-life adversity, including maternal loss, on children's HPA axis functioning. Adults up to 64 years old who experienced mistreatment and/or the loss of one or both parents during childhood have, depending on the study, a lower (*Meinlschmidt and Heim, 2005*; *Kawai et al., 2017*) or higher cortisol-awakening response (*Gonzalez et al., 2009*; *Butler et al., 2017*), flatter diurnal cortisol slopes (*Karlamangla et al., 2019*), and generally higher cortisol levels throughout the day (*Nicolson, 2004*). However, the effects of parental loss on HPA axis activity are not long-lasting under every condition. In support of the ACM, human orphans adopted at an early age presented diurnal cortisol excretion profiles comparable to children raised by their biological parents (*Gunnar et al., 2001*). In contrast, orphans placed for extended periods in orphanages in which they were exposed to drastic food restriction and physical violence presented higher cortisol levels than mother-reared children, particularly in the evening (*Gunnar et al., 2001*).

Studies on captive non-human primates also reveal that the social environment can mediate the physiological effects of maternal loss. Orphan bonobo immatures raised in sanctuaries by surrogate human mothers presented similar cortisol levels to mother-reared immatures (*Wobber and Hare, 2011*), whereas nursery-reared orphan macaques, without surrogate mothers, presented blunted morning cortisol levels (*Thomas et al., 1995*). These studies on human and non-human primates highlight the flexible nature of the HPA axis functioning and how it can be reshaped by changes in the environment throughout development. However, the social support available to orphans in these captive studies is distinct from the potential support of conspecifics in the wild. Therefore, evaluating both the BEM and the ACM in wild long-lived animals with a slow life history is essential to understand the evolutionary roots of the human stress response and its role in developmental plasticity.

Wild long-lived mammals are adapted to the environment in which we typically observe them and in which selection may have favored mechanisms of rapid recovery from early-life traumatic events such as maternal loss to avoid long-term hyperactivation of the HPA axis (or chronic stress; *Beehner and Bergman, 2017*). Indeed, unlike humans, most long-lived mammalian species do not demonstrate alloparental care (*Lukas and Clutton-Brock, 2012*); therefore, in such species, individuals that lose their mothers during ontogeny may lack social support from their conspecifics following maternal loss.

A study on wild female baboons, using extensive long-term data, showed that simultaneous exposure to several forms of early-life adversity, and some isolated forms of adversity such as drought and low maternal rank, leads to an overall elevation in glucocorticoid levels in adulthood (*Rosenbaum et al., 2020*), offering support for the BEM. However, maternal loss in isolation did not lead to long-term elevation of glucocorticoid levels, suggesting that baboons may have buffering mechanisms to offset the effects of biological embedding for some forms of early-life adversity.

To our knowledge, this study on wild baboons constitutes the only test of the BEM in a wild long-lived non-human mammal. More studies are necessary to investigate if, or how extensively, the BEM and/or the ACM models apply to a wider range of long-lived species, both during development and in adulthood. In particular, it is important to test this model in long-lived species with a life history closer to that of humans. Baboons start reproducing 3–4 years after weaning, whereas humans and great apes, including chimpanzees, share an extended juvenile phase between weaning and first reproduction (*Wittig and Boesch, 2019a*). Furthermore, the study on baboons only assessed one marker of the HPA axis functioning (i.e., overall glucocorticoid levels) but did not investigate the impact on diurnal cortisol slopes. Assessments of diurnal cortisol slopes are important since these slopes are a marker of the HPA axis functioning (*Karlamangla et al., 2019*).

Using a long-term database, including demographic and urinary cortisol data, collected over a 19-year period on four wild Western chimpanzee communities (*Pan troglodytes verus*), we provide a rare test of the BEM and the ACM in a wild long-lived mammal. Specifically, our dataset allowed us to assess both the short- and long-term effects of maternal loss on the HPA axis activity in wild chimpanzees. We thereby investigated one of the potential physiological mechanisms explaining the fitness costs associated with maternal loss reported in wild chimpanzees such as reduced growth, survival, and reproductive success (*Nakamura et al., 2014*; *Samuni et al., 2020*; *Stanton et al., 2020*; *Crockford et al., 2020*). Furthermore, studying physiological effects using diurnal cortisol slopes is an underused paradigm in wild animal subjects despite its prevalence in the human health literature. In chimpanzees, these slopes are repeatable in adults (i.e., are consistent within a given individual over time, *Sonnweber et al., 2018*) but also show plasticity to physiological challenges such as disease outbreaks (*Behringer et al., 2020*) or aging (*Emery Thompson et al., 2020*).

For immatures, we investigated the effect of maternal loss in both sexes. For adult individuals, we focused on males, the philopatric sex in chimpanzees (*Pusey, 1979*; *Boesch and Boesch-Achermann, 2000*), since the early-life history of adult females who immigrated as adults into our study groups is often undocumented. For both age-class groups (male and female immatures and adult males), we first assessed whether average cortisol levels and the steepness of the diurnal cortisol slopes differed between orphan and non-orphaned individuals. We predicted that, as in humans, immature orphans would exhibit higher overall cortisol levels and flatter diurnal cortisol slopes than non-orphans. We also predicted that the overall effect of maternal loss on cortisol profiles would be more severe during the first years following maternal loss. That is because recently orphaned individuals have to adjust behaviorally and physiologically to a new social situation in which they do not benefit from maternal support and may have reduced access to food and socio-positive social interactions. Over time, orphans may adjust and develop compensatory strategies for this adversity, which could result in lower impacts on the HPA axis activity. Wild chimpanzees are known to provide social support to orphaned individuals, ranging from tolerance in feeding sites to full adoptions (i.e., daily consistent provisioning of care to the orphans such as carrying, grooming, food sharing; *Uehara and Nyundo, 1983*; *Goodall, 1986*; *Wroblewski, 2008*; *Boesch et al., 2010*; *Hobaiter et al., 2014*; *Samuni et al., 2019a*). This may provide similar social buffering to that observed in some human populations or in captive primate studies. Accordingly, following a peak in the modification of the orphan cortisol profile directly after maternal loss, we anticipated, as would be predicted by the ACM, some decline over time, but still for cortisol levels to remain elevated in orphans compared to non-orphans. Since the life history of chimpanzees, and especially the extended immature phase, resembles more the life history of humans than that of baboons, we predicted this to last even into adulthood, matching the patterns of long-lasting HPA activity alteration arising from maternal loss in humans.

Finally, in line with the ACM, which predicts differential adaptive physiological response at different stages of ontogeny, we predicted that immature orphans that lost their mothers earlier in their lives would have flatter diurnal cortisol slopes and overall higher cortisol levels than immatures who lost their mother at a later age due to a greater level of dependency on mothers in early ontogeny (*Clark, 1977*; *Pusey, 1983*; *Boesch and Boesch-Achermann, 2000*).

## Results

We used the long-term behavioral, demographic, and urine sample data of the Taï Chimpanzee Project (*Wittig and Boesch, 2019b*) collected on four communities of wild Western chimpanzees (East, North, Middle, and South) in the Taï National Park, Cote d'Ivoire (5°52′N, 7°20′E). The urine samples included in this study span over 19 years and were collected between 2000 and 2018 (see *Table 1* for details about sample size).

We used a series of Bayesian linear mixed models (LMMs) to test our predictions regarding the effect of maternal loss on overall cortisol levels and diurnal slopes (jointly constituting the cortisol profile). We first tested these effects in socially immatures (i.e., males and females < 12 years of age because prior to 12 years, chimpanzees associate primarily with their mother; *Reddy and Sandel, 2020*). Secondly, we tested these effects in mature males (i.e., males ≥ 12 years of age).

For immatures, we first tested for differences in cortisol profiles between recently orphaned individuals (individuals that lost their mother for less than 2 years previously, N = 7), not-recently orphaned orphans (individuals that experienced maternal loss more than 2 years ago at the time of sampling,

**Table 1.** Sample size for immature and adult male orphans and non-orphans in each of the four study communities.

| Community | Age class | Orphan status | N. ID | No. of samples | Mean ± SE no. of sample per individual | Age range (years) |
|---|---|---|---|---|---|---|
| | | Non-orphans | 8* | 112 | 14.0 ± 3.8 | 3.8–11.9 |
| | Immatures | Recently orphaned | 5* | 116 | 23.2 ± 9.1 | 4.1–11.9 |
| | | Non-recently orphaned | 9* | 136 | 15.1 ± 6.4 | 6.1–11.8 |
| | | Non-orphans | 4 | 456 | 114.0 ± 56.2 | 13.8–40.7 |
| Taï East | Adult males | Orphans | 3 | 354 | 118.0 ± 18.0 | 12.3–19.8 |
| | | Non-orphans | | | | |
| | Immatures | Recently orphaned | | | | |
| | | Non-recently orphaned | | | | |
| | | Non-orphans | 3 | 17 | 5.7 ± 0.67 | 17.2–33.7 |
| Taï Middle | Adult males | Orphans | | | | |
| | | Non-orphans | 11 | 168 | 15.3 ± 4.3 | 3.0–12.0 |
| | Immatures | Recently orphaned | | | | |
| | | Non-recently orphaned | 2 | 7 | 3.5 ± 0.5 | 10.5–11.3 |
| | | Non-orphans | 3 | 95 | 31.7 ± 9.1 | 12.3–20.8 |
| Taï North | Adult males | Orphans | 2 | 61 | 30.5 ± 3.5 | 12.1–20.4 |
| Taï South | | Non-orphans | 17* | 173 | 10.2 ± 1.5 | 2.8–11.9 |
| | Immatures | Recently orphaned | 2* | 18 | 9.0 ± 6.0 | 4.1–9.2 |
| | | Non-recently orphaned | 5* | 116 | 23.2 ± 7.7 | 6.1–11.9 |
| | Adult males | Non-orphans | 7 | 858 | 122.6 ± 18.1 | 12.1–45.3 |
| | | Orphans | 6 | 343 | 57.2 ± 31.5 | 12.1–21.9 |

*50 immature individuals were included in this study but one immature in Taï East and two immatures in Taï South were sampled before and after their mother died (i.e., they are counted twice in the table, once as an orphan and once as a non-orphan). Six males were included in the study as both mature and immature individuals.

N = 16), and non-orphans (N = 36; *all immature model*). We used these three categories to test whether the potential effect of maternal loss on immature cortisol profiles is a short-term effect (i.e., only present in recently orphaned individuals) or endures throughout ontogeny. Note that immatures that have been sampled over several years can appear in two or three of the orphan categories in our dataset. We then tested, for immature orphans only (N = 17), the effect of two test predictors on their cortisol profiles, the age at which the orphan lost their mother, and the years since maternal loss (*immature orphan model*).

For mature males, we tested first for differences in cortisol profiles between orphans (individuals who were orphaned before reaching 12 years of age, N = 11) and non-orphans (N = 17, *all adult male model*). We then tested, for orphan mature males only (N = 11), the effect of the age at which the orphans lost their mother on their cortisol profiles (*adult male orphan model*). In all the models, each urine sample represented a data point, and the log-transformed cortisol concentration of the sample (expressed in ng/ml SG) was the response variable. We summarize below the test predictors we used in each model:

- In the *all immature model*, we used as a test predictor a categorical variable for *orphan status* with three levels: recently orphaned (within 2 years after maternal loss), not-recently orphaned (longer than 2 years since maternal loss), and non-orphans.
- In the *immature orphan model*, we used two test predictors: *age when mother died,* a continuous variable describing the age at which immature orphans lost their mother and *years since maternal loss,* a continuous variable describing the number of years since immature orphans lost their mother.
- In the *all adult male model*, we used as a test predictor a binary variable for *orphan status*, namely if the mature male had been orphaned before reaching 12 years of age yes/no.
- In the *adult male orphan model*, we used as a test predictor *age when mother died,* a continuous variable as in the *immature orphan model*.

For each of the models, all these test predictors were included in interaction with the linear, the quadratic, and the cubic terms for time of sample collection to test the effect of these test predictors on diurnal cortisol slopes. The quadratic and cubic terms for time of sample collection were included here since a previous study using a large sample size showed that diurnal cortisol slopes in chimpanzees follow a curved cubic pattern (*Emery Thompson et al., 2020*). While our models differ in the test predictors that were included in each model, the control predictors were nearly identical for all models (except for the *immature orphan model*, see below). In all our models, we controlled for sex of the individual, community size, sex ratio of mature individuals in the community, age of the individual (except for the *immature orphan model* because of collinearity issue, see Materials and methods), the liquid chromatography mass spectrometry (LCMS) method used ('old' or 'new' method, see the Urine analysis section), and seasonal variation in ecological conditions (see Materials and methods). In addition, we controlled for repeated observations of the same individual over the same year by incorporating *individual ID* and *year* as random factors in each model. Finally, to control for the changes in cortisol diurnal slope with age, we built one slope per individual per year into each model by incorporating the dummy variable *individual_year* as a random factor. For each model, we first ran a full model comprising all the variables and interactions described above. When the 90% credible interval (CI) for the estimate of an interaction term overlapped 0, this indicated a large uncertainty and therefore that the effect of this interaction was not consistent in our data. Accordingly, we reran the models without those non-consistent interaction terms and present the results of the final reduced models. For each model, we report *P* + and *P*- as the percentage of the posterior distribution in support of the hypothesized positive or negative effect given the observed data. We also report the proportion of variance in the response explained by the fixed effects and the random effects (conditional $R^2$) and the fixed effect only (marginal $R^2$). Throughout the result section, we describe differences in cortisol levels in the morning, in the afternoon, or throughout the day, based on the visual inspection of the model line predictions and on the model output regarding diurnal cortisol slopes. We did not directly test differences in cortisol levels in morning or afternoon samples separately due to limitations in the sample size, but these differences can be assessed based on the diurnal cortisol slopes depicted in the figures.

## Effect of maternal loss on immature cortisol profiles

For the *all immature model* (N = 846 samples and 50 individuals), assessing if immature individuals that were either recently orphaned, not-recently orphaned, and non-orphans differed in their cortisol profiles, the 90% CI for the estimate of the interaction terms between orphan status and the quadratic and the cubic term for *time of the day* overlapped with 0. In the reduced model not comprising these two interaction terms, we found that recently orphaned individuals had a consistently steeper linear slope (estimate for the interaction between orphan status and the linear term for *time of the day*: –0.22, 95% CI: [–0.03: –0.48], P = 98.9%, *Table 2*, *Figure 1*, and *Appendix 1—figure 1*) than non-orphans. On average, recently orphaned individuals had a diurnal cortisol slope 58% steeper than non-orphans (average slope and [95% CI] for recently orphaned: –0.60 [-0.79: –0.41], and non-orphans: –0.38 [-0.53: –0.22]). A visual inspection of the model line prediction (*Figure 1*) indicates that this difference in slopes may stem from higher early morning cortisol levels in recently orphaned immatures as compared to non-orphan immatures (*Figure 1* and *Appendix 1—figure 1*). However, we found no consistent differences in the linear slopes of not-recently orphaned immatures and non-orphan immatures (90% CI: [–0.24: 0.01], P = 93.4%, *Table 2*, *Figure 1*, and *Appendix 1—figure 1*). Since our model comprised an interaction between orphan status and time of the day, the main effect

**Table 2.** Results of the all immature and the immature orphan models.

These two models tested in immatures for the effect of orphan status (all immature model) and, for immature orphan only, the effect of age when mother died and years since maternal loss (orphan immature model) on cortisol levels and diurnal cortisol slopes. The results presented here are from reduced all immature and immature orphan models after removing the interactions for which the 90% credible interval (CI) overlapped 0. SE indicates the standard error of the estimate for each predictor. The coded level for each categorical predictor is indicated in parentheses. Control predictors are italicized. 95% CI low and 95% CI high indicate the lower and upper limits of the 95% CI. Likewise, 90% CI low and 90% CI high indicate the lower and upper limits of the 90% CI. CIs that do not overlap 0 are indicated in bold. Predictors for which the 95% CI did not overlap 0 are indicated with salmon shading, and predictors for which the 95% CI overlapped 0 but the 90% CI did not overlap 0 are indicated with yellow shading. LCMS: liquid chromatography mass spectrometry. Time of the day$^2$: quadratic term for time of the day. Time of the day$^3$: cubic term for time of the day.

| Model | Response | Predictor | Estimate | SE | 95% CI low | 95% CI high | 90% CI low | 90% CI high |
|---|---|---|---|---|---|---|---|---|
| All immature model | Log urinary cortisol levels (ng/ml SG) | Intercept | 3.69 | 0.27 | 3.17 | 4.23 | 3.26 | 4.14 |
| | | Time of the day | –0.38 | 0.08 | **–0.53** | **–0.22** | **–0.51** | **–0.25** |
| | | Time of the day$^2$ | –0.01 | 0.08 | –0.17 | 0.14 | –0.14 | 0.11 |
| | | Time of the day$^3$ | 0.03 | 0.04 | –0.05 | 0.12 | –0.04 | 0.10 |
| | | Orphan category (orphan for less than 2 years) | 0.01 | 0.19 | –0.36 | 0.38 | –0.30 | 0.32 |
| | | Orphan category (orphan for more than 2 years) | –0.14 | 0.22 | –0.58 | 0.32 | –0.50 | 0.23 |
| | | *Sex ratio* | 0.07 | 0.07 | –0.06 | 0.2 | –0.04 | 0.18 |
| | | *Community size* | –0.12 | 0.06 | –0.24 | 0.01 | **–0.22** | **–0.01** |
| | | *Individual sex (male)* | –0.26 | 0.25 | –0.78 | 0.22 | –0.67 | 0.13 |
| | | *Individual age at sample* | –0.01 | 0.17 | –0.33 | 0.37 | –0.27 | 0.29 |
| | | *LCMS method (old)* | 0.06 | 0.35 | –0.62 | 0.78 | –0.50 | 0.64 |
| | | *Sin(seasonDate)* | –0.07 | 0.04 | –0.14 | 0 | –0.13 | –0.02 |
| | | *Cos(seasonDate)* | 0.01 | 0.04 | –0.06 | 0.08 | –0.05 | 0.07 |
| | | Orphan category (less than 2 years): time of the day | –0.22 | 0.10 | **–0.41** | **–0.03** | **–0.38** | **–0.07** |
| | | Orphan category (more than 2 years): time of the day | –0.11 | 0.08 | –0.27 | 0.03 | –0.24 | 0.01 |

*Table 2 continued on next page*

*Table 2 continued*

| Model | Response | Predictor | Estimate | SE | 95% CI low | 95% CI high | 90% CI low | 90% CI high |
|---|---|---|---|---|---|---|---|---|
| Immature orphan model | Log urinary cortisol levels (ng/ml SG) | Intercept | 3.59 | 0.26 | 3.08 | 4.11 | 3.17 | 4.02 |
| | | Time of the day | –0.44 | 0.17 | –0.75 | –0.05 | –0.7 | –0.14 |
| | | Time of the day$^2$ | –0.01 | 0.07 | –0.16 | 0.13 | –0.13 | 0.1 |
| | | Time of the day$^3$ | 0.03 | 0.11 | –0.18 | 0.27 | –0.14 | 0.22 |
| | | Orphan's age when mother died | 0.00 | 0.24 | –0.51 | 0.46 | –0.41 | 0.38 |
| | | Years since maternal loss | –0.16 | 0.16 | –0.48 | 0.14 | –0.42 | 0.09 |
| | | Sex ratio | –0.06 | 0.11 | –0.27 | 0.15 | –0.23 | 0.11 |
| | | Community size | –0.05 | 0.08 | –0.22 | 0.12 | –0.19 | 0.09 |
| | | Individual sex (male) | 0.1 | 0.3 | –0.47 | 0.7 | –0.38 | 0.6 |
| | | LCMS method (old) | –0.04 | 0.32 | –0.66 | 0.64 | –0.55 | 0.51 |
| | | Sin(seasonDate) | 0.02 | 0.05 | –0.09 | 0.12 | –0.07 | 0.1 |
| | | Cos(seasonDate) | 0.04 | 0.05 | –0.07 | 0.14 | –0.05 | 0.12 |
| | | Years since maternal loss: time of the day | 0.19 | 0.1 | –0.01 | 0.38 | 0.02 | 0.35 |
| | | Years since maternal loss: time of the day$^2$ | –0.05 | 0.05 | –0.15 | 0.05 | –0.14 | 0.03 |
| | | Years since maternal loss: time of the day$^3$ | –0.06 | 0.04 | –0.15 | 0.02 | –0.14 | 0.00 |
| | | Age when mother died: time of the day | 0.08 | 0.09 | –0.09 | 0.25 | –0.06 | 0.22 |
| | | Age when mother died: time of the day$^2$ | –0.12 | 0.06 | –0.23 | –0.00 | –0.21 | –0.02 |

of orphan status inherently corresponds only to the difference at midday. The 95% CI for the main effect of orphan status in the *all immature model* clearly overlapped 0, indicating that there was no consistent difference between recently orphaned, not-recently orphaned, and non-orphans in their average cortisol levels at midday. However, *Figure 1* indicates that recently orphaned individuals had higher cortisol levels in the morning and lower cortisol levels in the afternoon than non-orphans. Conditional and marginal $R^2$ for the *all immature model* were 0.60 and 0.26, respectively.

The second model (*immature orphan model*, N = 393 samples and 17 individuals) focusing on immature orphans revealed that orphans varied in their cortisol profiles depending on the age at which their mother died and on the length of time since they were orphaned. More specifically, and in line with the results of the *all immature model*, immature orphans whose mother had died several years before sampling had different diurnal cortisol slopes as compared to more recently orphaned immatures (estimate and [90% CI] for the interaction between *years since maternal loss* and the cubic term for *time of day*: –0.06 [-0.14: 0.00]). However, this difference comprised a relatively large uncertainty since the 90% CI comprised 0, the 95% CI overlapped 0 (*Table 2*), and *P-* = 93.8%. Immature orphans who lost their mother recently had a diurnal cortisol slope that curved upwards in the afternoon (orange squares in *Figure 2*). A visual inspection of the data and the model prediction lines reveals that cortisol levels of immature orphans who recently lost their mother had higher cortisol levels throughout the day (orange squares in *Figure 2*) as compared to immature orphans who lost their mother several years ago (green triangles in *Figure 2*). The difference in cortisol levels between recently and non-recently orphaned individuals was most evident during early morning and late afternoon (*Figure 2*).

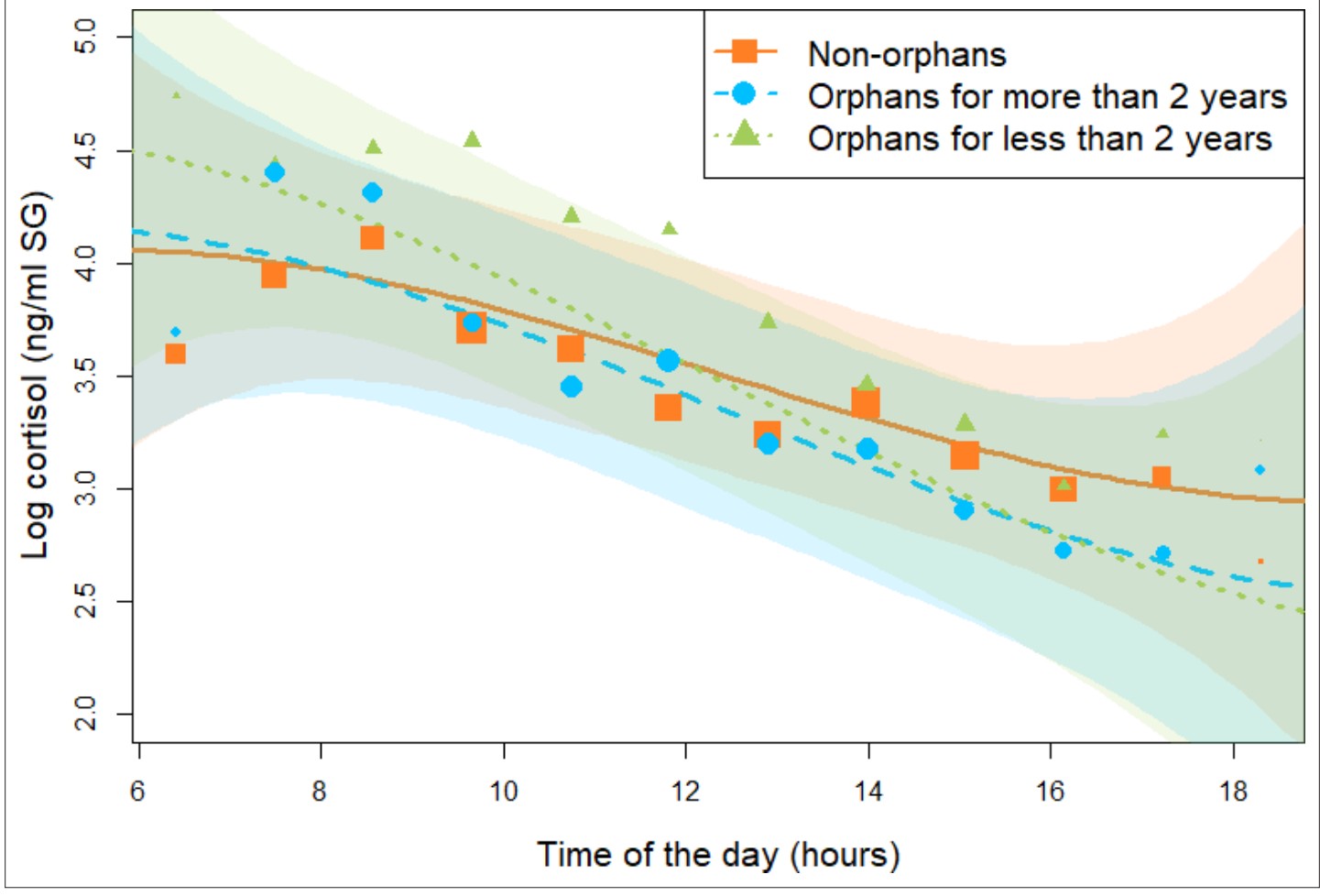

**Figure 1.** Effect of maternal loss on daily urinary cortisol level variations in immature chimpanzees. Non-orphans are depicted by orange squares, recently orphans (orphaned for less than 2 years) by green triangles, `and non-recently orphaned (orphaned for more than 2 years) by blue circles. Each dot represents the average hourly cortisol level for all individuals of each orphan category. The size of the dot is proportional to the sample size (e.g., the number of data points) contributing to each dot. The orange solid line and green and blue dashed lines depict the model line prediction, and the orange, green, and blue area the 95% credible interval (CI) from the *all immature model* for non-orphans, recently orphans, and non-recently orphans, respectively. The model lines depict the consistent effect of the interaction between *orphan status* and the linear term for time of day in the *all immature model* (estimate: –0.22, 95% CI: [–0.03: –0.48]). The sample size for the *all immature model* was N = 846 samples and 50 individuals.

In addition, in the *immature orphan model*, we found a consistent effect of the interaction between *age when mother died* and the quadratic term of *time of day* (estimate and [95% CI]: –0.12 [-0.23: –0.00], *P-* = 97.8%, *Table 2*), indicating that the age at which immatures lost their mother consistently influenced their diurnal cortisol slopes and more specifically how it curved throughout the day. Immature individuals who lost their mother before 5 years of age (orange squares in *Figure 3*) had a diurnal cortisol slope that curved upwards. A visual inspection of the model prediction lines and the data points reveals that immatures orphaned at a younger age presented higher early morning and late afternoon cortisol levels than individuals who lost their mother at an older age. Those who lost their mother between 5 and 8 years of age (blue circles in *Figure 3*) had a relatively linear decrease in cortisol levels throughout the day. Finally, immature individuals orphaned between 8 and 12 years of age (green triangles in *Figure 3*) had a diurnal cortisol slope that curved downwards with lower early morning and late afternoon cortisol levels than individuals who lost their mother at a younger age. Please note that we depicted model line predictions for three age categories corresponding to three life history stages in chimpanzees (below 5 years: infancy, 5–8 years: juvenile period, 8–12 years: early adolescence) in *Figure 3* for ease of interpretation of the effect, but the variable 'age when mother

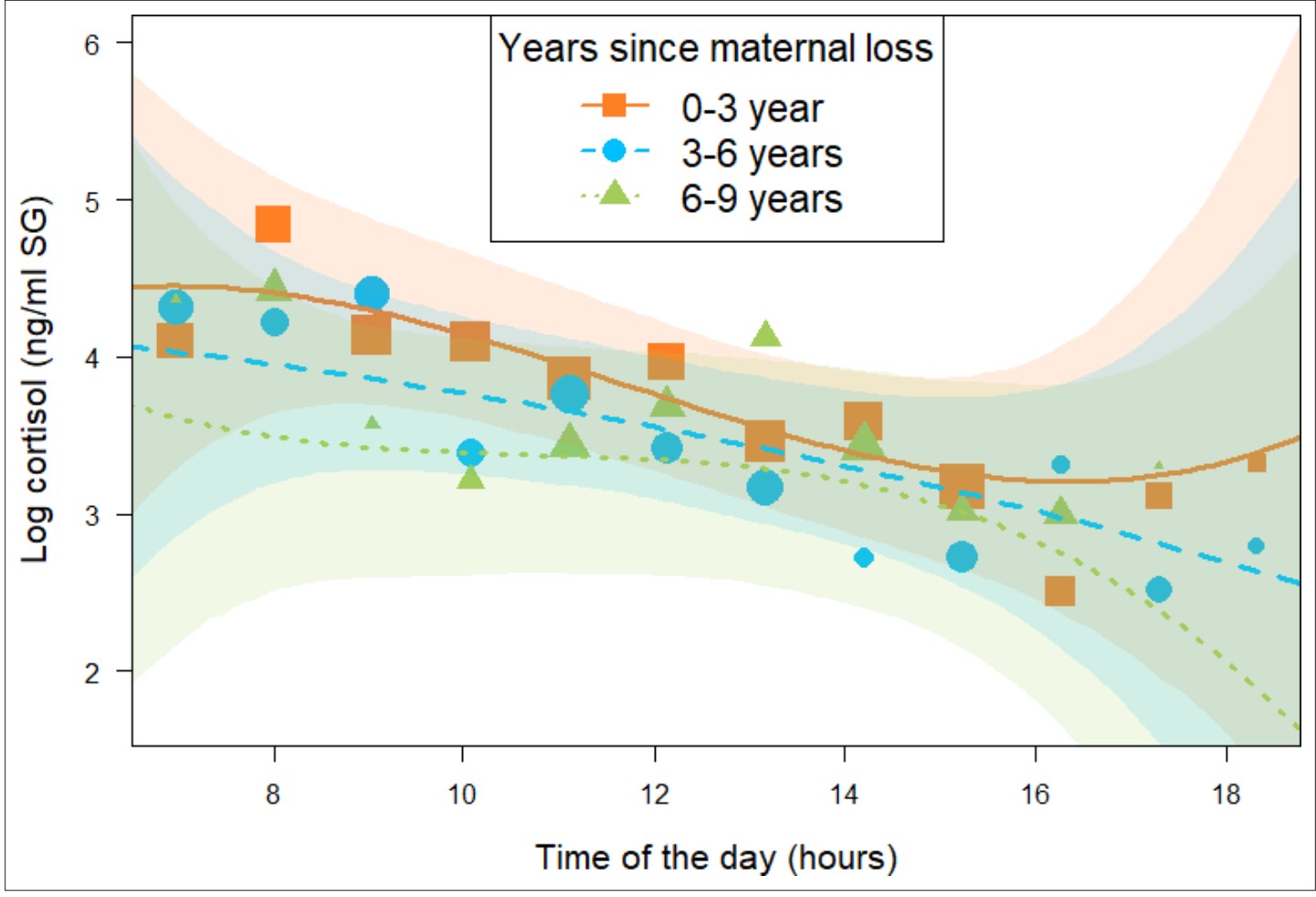

**Figure 2.** Effect of time (in years) since being orphaned on daily urinary cortisol level variations of immature orphan chimpanzees. Orange squares, blue circles, and green triangles depict cortisol levels for individuals who lost their mother less than 3 years ago, 3–6 years ago, and 6–9 years ago, respectively. Each dot represents the average hourly cortisol level of all individuals for each of the three categorical intervals of years since maternal loss. The size of the dot is proportional to the sample size (number of urine samples collected) for each hour of the day. The orange solid line, blue dashed line, and green dashed lines indicate model line predictions, and the orange, blue, and green light areas indicate the 95% credible interval (CI) for individuals who lost their mother less than 3 years ago, 3–6 years ago, and 6–9 years ago, respectively (*immature orphan model*). The model line depicts the interaction between *years since maternal loss* and the cubic term for *time of day* in the *immature orphan model*: (estimate and [90% CI]: –0.06 [-0.14: 0.00]), but this effect comprised a relatively large uncertainty since the 95% CI overlapped 0 (Table 2). *Note that while the predictor 'years since maternal loss' was modeled as a continuous variable in the immature orphan model, for ease of interpretation we depict the results here for three categorical intervals of 'years since maternal loss'.* The sample size for the *immature orphan model* was N = 393 samples and 17 individuals.

died' was incorporated in the *immature orphan model* as a continuous predictor. Conditional and marginal $R^2$ for the *immature orphan model* were 0.64 and 0.32, respectively.

## Effect of maternal loss on cortisol slopes in mature male chimpanzees

In contrast to immature individuals, we did not detect a consistent effect of *orphan status* on mature males' diurnal cortisol profiles and cortisol levels. All the 90% CIs for the estimate of the interaction terms between orphan status and the linear, quadratic, and cubic terms for time of day in the full *all adult male model* largely overlapped 0, and all *P+* and *P-* were below 75% for the estimates of each interaction (N = 2184 samples and 28 individuals; *Appendix 1—table 1*). Furthermore, in a reduced model not comprising these interactions, the 90% CI for the estimate of orphan status largely overlapped 0 and *P+* = 56% (*Appendix 1—table 1*). Conditional and marginal $R^2$ for the *all adult male model* were 0.53 and 0.28, respectively.

We also did not detect a consistent effect of the *age when mother died* on orphan mature males' diurnal cortisol slopes and cortisol levels. All the 90% CIs for the estimates of the interaction terms

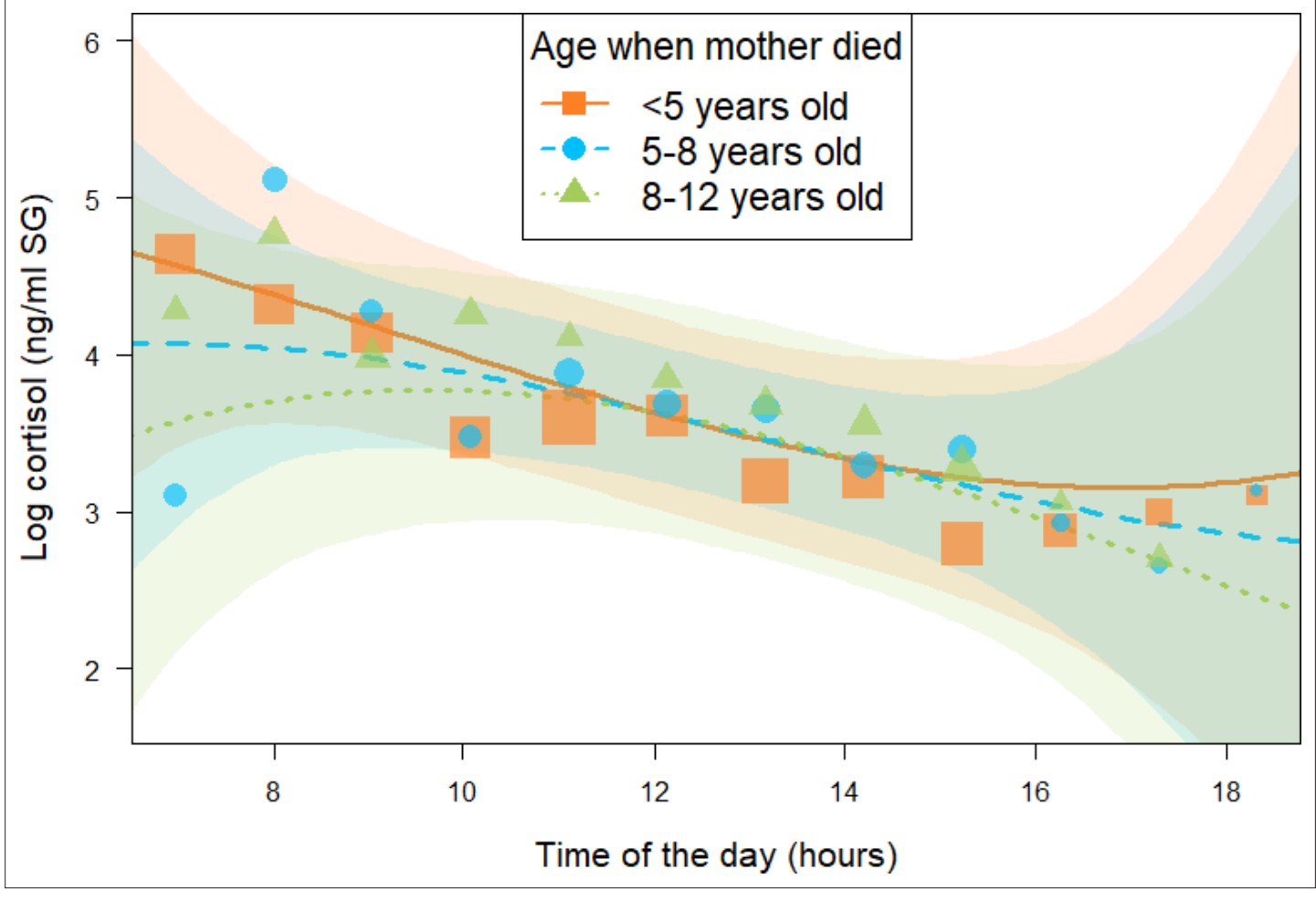

**Figure 3.** Effect of age at which immature orphans lost their mother on daily urinary cortisol level variations of immature orphan chimpanzees. Orange square, blue circles, and green triangles depict cortisol levels for individuals who lost their mother when they were less than 5 years, 5–8 years, and 8–12 years of age, respectively. Each dot represents the average hourly cortisol level of all individuals for each of the three categories of age when mother died. The size of the dot is proportional to the sample size (number of urine sample collected) for each hour of the day. The orange solid line and blue and green dashed lines indicate model line predictions, and the orange, blue, and green light areas the 95% credible interval (CI) for individuals who lost their mother when they were less than 5 years, 5–8 years, and 8–12 years of age, respectively (*immature orphan model*). The model lines depict the consistent effect of the interaction between *age when mother died* and the quadratic term of *time of day* in the *immature orphan model* (estimate and [95% CI]: –0.12 [0.00: –0.23]). *Note that while 'age at maternal loss' was modeled as a continuous variable in the immature orphan model, for ease of interpretation the model is depicted here for three categorical intervals of 'age when mother died'.* The sample size for the *immature orphan model* was N = 393 samples and 17 individuals.

between *age when mother died* and the linear, quadratic, and cubic terms for *time of day* in the full *adult male orphan model* overlapped 0, and all *P + and P-* were below 92% (N = 769 samples and 10 individuals; *Appendix 1—table 2*). In a reduced model not comprising these interactions, the 90% CI for the estimate of *age when mother died* largely overlapped 0 and *P- = 62%* (*Appendix 1—table 2*). Conditional and marginal $R^2$ for the *adult male orphan model* were 0.60 and 0.38, respectively.

## Discussion

While the effect of maternal loss on wild animal survival and reproduction has been recently established (*Foster et al., 2012*; *Andres et al., 2013*; *Tung et al., 2016*; *Walker et al., 2018*; *Surbeck et al., 2019*; *Crockford et al., 2020*; *Zipple et al., 2021*), the mechanisms underlying these fitness costs remain understudied. Our study provides one of the rare empirical tests of the BEM (see also *Rosenbaum et al., 2020*) and ACM in wild long-lived mammals by assessing the short- and long-term physiological impacts of early maternal loss. While we found an effect of maternal loss on diurnal

cortisol slopes in immature chimpanzees whose mothers died recently (*all immature* and *immature orphan models*), these effects were neither present in individuals who lost their mothers more than 2 years earlier (*all immature* and *immature orphan models*) nor in mature male chimpanzees (*adult male orphan model*). These results are in line with the absence of long-term effects of maternal loss alone on glucocorticoid levels in wild long-lived baboons (*Rosenbaum et al., 2020*). This suggests that the BEM (*Power and Hertzman, 1997*; *Miller et al., 2011*; *Berens et al., 2017*) may apply to long-lived wild mammals only following exposure to a combination of diverse sources of early-life adversity or to other sources of early-life adversity than maternal loss (see *Rosenbaum et al., 2020*). Our results also provide tentative support for the ACM. The ACM predicts that exposure to adversity leads to modification of the HPA axis activity, but that these modifications are more or less long-lasting depending on the social and ecological environment faced by the developing individual after exposure to adversity (*Del Giudice et al., 2011*). In the case of our study, amelioration in the social environment of immature chimpanzees, possibly in the form of buffering mechanisms ranging from minimal alloparental care to adoption (discussed below), could modify the HPA axis activity of orphans throughout ontogeny and eventually ameliorate long-term effects of early-life adversity. Alternatively, in our study, a survivorship bias may mean that chimpanzees with severely altered HPA axis activity following maternal loss did not survive to adulthood and thus were absent from our adult dataset. We also found that the age at maternal loss impacted the diurnal cortisol slopes of the orphans (*immature orphan model*). Orphans experiencing maternal loss at younger ages had a diurnal cortisol slope differing from the immatures orphaned when older, particularly in the quadratic term for time of the day (i.e., in how the slope curved). Orphans that lost their mother at younger ages had diurnal cortisol slopes that curved upwards towards the end of the day, whereas those that lost their mother later in ontogeny showed a more typical pattern observed in non-orphan immatures of more continual declines in their slopes. Visual inspection of the model lines and data points revealed that this difference in the slope led to higher early morning and late afternoon cortisol levels in orphan whom mother died early than in those of immatures orphaned when older. This latter finding is also in line with the ACM by highlighting potentially different physiological adaptive responses at different stages of ontogeny. In fact, diurnal cortisol slopes are indicative of the general functioning of the HPA axis (*Karlamangla et al., 2019*) and our results indicate that the diurnal cortisol slope of immature chimpanzees undergoes different levels of changes depending on the age at which they experience maternal loss, with more substantial deviations from mother-raised offspring pattern in immatures orphaned earlier in life.

In line with our prediction, orphans had different diurnal cortisol slopes as compared to non-orphans (*all immature model*). However, this difference was only present for orphans who recently lost their mother, that is, within 2 years at the time of sampling. Furthermore, the direction of the effect was opposite to the prediction derived from the clinical human literature (*Kaufman, 1991*; *Hart et al., 1996*; *Meinlschmidt and Heim, 2005*; *Dozier et al., 2006*; *Bernard et al., 2015*; *McLachlan et al., 2016*) in that recently orphaned immatures had a steeper rather than a flatter diurnal cortisol slopes compared to those of non-orphaned immatures. An inspection of the raw data (*Figure 1* and *Appendix 1—figure 1*) reveals that this steeper slope is more likely to be driven by higher early morning cortisol levels rather than by lower late afternoon cortisol levels.

In humans, flatter diurnal cortisol slopes in individuals who experienced early-life adversity are often related to higher afternoon cortisol levels (*Carlson and Earls, 1997*; *Gunnar et al., 2001*; *Tarullo and Gunnar, 2006*). A failure to bring cortisol levels down in the afternoon has been interpreted as a fundamental dysregulation within the neuroendocrine system (*Young et al., 1994*). Such dysregulation might not apply to recently orphaned immature chimpanzees since, as compared to non-orphans, their diurnal cortisol slopes were steeper and their afternoon cortisol levels were not necessarily higher. The diurnal cortisol profile of recently orphaned immature chimpanzees better mirrors the cortisol profiles of individuals exposed to nutritional stress. Studies on humans and captive rats show that dietary restriction is associated with a rise in early morning cortisol levels (*Goodwin et al., 1988*; *Garcia-Belenguer et al., 1993*). Nutritional stress in orphan chimpanzees may result from the lack of access to food sources that were initially provided by the mother (*Pusey, 1983*; *Goldenberg and Wittemyer, 2017*; *Samuni et al., 2019a*). Although chimpanzees in our study were orphaned after weaning age (i.e., after 4 years of age, *Samuni et al., 2020*; *Appendix 1—table 3*), these orphans may nonetheless be constrained in acquiring the amount of food necessary to maintain a positive energy balance. This may be because orphans lack socially facilitated access to food sources

previously provided by their mother and also miss out on maternal sharing of high nutrient food, such as meat, nuts, and honey, which occurs intermittently throughout ontogeny (*Samuni et al., 2019a*). During the period directly following the loss of their mother, orphan chimpanzees might be exposed to acute stress, which modifies adaptively their HPA axis activity, allowing maintenance of homeostasis while developing and eventually survival until maturity, despite the potential lack of energy intake.

Such modifications, and in particular early morning cortisol elevations, may mediate other fitness-related traits. For instance, in the same population, orphans lose out on growth compared to non-orphans (*Samuni et al., 2020*), which might result from reallocation of energy towards more vital functions (e.g., thermoregulation, muscle activity, general organismal functioning) than growth during energetically challenging periods. Early morning cortisol levels can be indicative of such re-allocation. In contrast, nutritional stress might be less prominent in sampled human orphans, which are predominantly studied in Western countries, where, even in orphanages, children are likely to receive sufficient amounts of food. This might explain why, unlike human orphans that have lower morning cortisol levels (*Meinlschmidt and Heim, 2005*; *Dozier et al., 2006*), recently orphaned chimpanzees being under nutritional stress have higher early morning cortisol levels than non-orphans.

The results of our orphan/non-orphan comparison in immature chimpanzees (*all immature model*) were partially confirmed by our second analysis that included only samples from immature orphans (*immature orphan model*). This allowed us to assess the effect of *years since maternal loss* as a continuous variable while accounting for the age at which each orphan lost their mother. In this analysis, we found that recently orphaned immatures and immatures who lost their mother at a younger age had higher early morning cortisol levels and higher late afternoon cortisol levels as compared to other orphans. However, we note that there was some uncertainty in the strength and direction of this effect, which is likely due to the limited sample size for this analysis and the variable nature of experiences among orphans following maternal loss (see below). Nevertheless, this result implies that recently orphaned immatures and orphans who lost their mothers at a younger age are exposed to more stressors during the time when chimpanzees are the most socially active (i.e., early morning and late afternoon). This could reflect higher nutritional or energetic stress for orphans when groups are most active. These early morning and late afternoon peaks could equally reflect exposure to social stressors when other individuals are most active. Play bouts involving orphan chimpanzees are shorter and more frequently escalate into aggression when compared to play bouts between non-orphans (*Botero et al., 2013*; *Leeuwen et al., 2014*). This could, in turn, lead to increased cortisol levels since in chimpanzees and other primates aggression generally increases cortisol levels (*Girard-Buttoz et al., 2009*; *Emery Thompson et al., 2010*; *Wittig et al., 2015*; but see *Preis et al., 2019*).

Mothers are the main social partners of immature chimpanzees *Reddy and Sandel, 2020*; therefore, orphan chimpanzees lack access to a key source of social buffering from environmental stressors (*Young et al., 2014*; *Wittig et al., 2016*). Orphans may later form relationships that provide such buffering, but in the short term, new orphans and/or those who lost their mother at a young age may be more exposed to cumulative social and psychological stressors without access to social buffering mechanisms. The social factors affecting chimpanzee orphans may be similar to those impacting human orphans who are frequent targets of assault due to a lack of social support from a caregiver (e.g., *Frank et al., 1996*).

As in humans, maternal loss impacts the physiology of immature chimpanzees. However, the effects of maternal loss on chimpanzees' physiology differ strongly from that of humans, in that they do not persist into adulthood. More strikingly, immatures that were orphaned for more than 2 years had cortisol excretion profiles that were not consistently different from the profiles of non-orphans. Adult humans up to 64 years old, which experienced mistreatment and/or the loss of one or both parents during childhood, still present alteration of their diurnal cortisol slopes and overall cortisol levels (*Nicolson, 2004*; *Meinlschmidt and Heim, 2005*; *Gonzalez et al., 2009*; *Kawai et al., 2017*; *Butler et al., 2017*; *Karlamangla et al., 2019*). The re-establishment of a normal functioning of the HPA axis in mature male chimpanzees but also in immatures orphaned for more than 2 years may reflect a form of recovery in those individuals. In turn, this potential recovery is in line with the ACM, which predicts that the HPA axis activity has been selected to be adaptively flexible throughout ontogeny, allowing the individuals to readjust HPA axis activity following critical developmental challenges (in the case of our study, maternal loss). These changes in HPA axis activity are conditional adaptations to match the social and physical environment (*Del Giudice et al., 2011*). The lack of apparent long-term effects

of maternal loss on immature chimpanzee physiology could thus be indicative of ameliorations in the environment of these orphans in the years following maternal loss, possibly in terms of improved access to social support and food.

Adoption by conspecifics is one such environmental modification that, in humans, is associated with cortisol profiles returning to similar levels as those of non-orphans (**Gunnar et al., 2001**). In chimpanzees, adoption is a relatively common phenomenon (**Uehara and Nyundo, 1983**; **Goodall, 1986**; **Wroblewski, 2008**; **Boesch et al., 2010**; **Hobaiter et al., 2014**; **Samuni et al., 2019a**). Adopters typically provide surrogate care for the orphan in the form of grooming, limited provisioning through food sharing, agonistic support, nest sharing, and even carrying (**Samuni et al., 2019a**). As a result, adoption may also increase the survival probability of the orphan (**Hobaiter et al., 2014**). It is possible that the adoption of immature orphan chimpanzees also alleviates some of the effects of maternal loss on cortisol excretion profiles as observed in humans (**Gunnar et al., 2001**). Unfortunately, data were insufficient to evaluate effectively the effect of adoption on cortisol excretion profiles in this sample (adoption status is available for 9 out of 17 orphans; **Appendix 1—table 3**). Most orphans for whom we have data were adopted at some point during their immature years (**Appendix 1—table 3**), but the level of care provided by the adopter varies greatly across orphans (**Samuni et al., 2019a**). Nevertheless, we lack detailed information on the alloparental care provided to most orphans in our sample to test the potential effect of the intensity of alloparental care on cortisol profiles.

While the ACM highlights that changes of the HPA axis activity during ontogeny when facing adversity are likely adaptive, it also emphasizes that exposure to extreme social situations, such as complete parental loss, may generate maladaptive phenotypes, especially if those events occur early during ontogeny (**Del Giudice et al., 2011**). Our results indicate that the age at maternal loss indeed has an impact on the orphan stress physiology in chimpanzees, but the subjects in our study may not have been orphaned at a young enough age to lead to long-term irreversible modifications of their physiology.

Long-term alteration of the HPA axis functioning related to early-life adversity is explained, at least partly, by epigenetic mechanisms (**Weaver et al., 2004**) and in particular by a hyper-methylation of DNA in regions coding for the glucocorticoid receptors (GRs) in the brain (**Liu et al., 1997**; **Weaver et al., 2007**, reviewed in **Zhang et al., 2013**). In rodents tested in laboratory conditions, these alterations of the GR in the brain take place early in life; a lower density of GR reduces the effectiveness of the glucocorticoid-negative feedback loop and ultimately results in prolonged elevated cortisol levels (**Zhang et al., 2013**). All orphan adult males in our study were orphaned after they were 4 years of age, an age at which the epigenetic effect on GR in the brain may be reduced or absent. Sampling individuals who lost their mother before weaning represents a methodological challenge since they often die soon after maternal loss (**Nakamura et al., 2014**; **Stanton et al., 2020**).

In conclusion, our study provides evidence of an effect of early-life adversity on diurnal cortisol slopes in immatures of a wild mammal population. Interestingly, our study provides contrasts to studies on humans by showing that the modifications of the HPA axis activity following maternal loss are not long-lasting. Even though modifications of the HPA axis activation may have some fitness consequences in wild mammalian populations (**Campos et al., 2021**), this mechanism is unlikely to explain the lower reproductive fitness and survival observed in adult male orphan chimpanzees (**Nakamura et al., 2014**; **Crockford et al., 2020**). In fact, in our study male orphans who reach adulthood do not present cortisol excretion profiles that differ from those of non-orphans. In wild baboons, maternal loss also did not have in itself long-term consequences on cortisol levels. Physiological embedding of maternal loss for this particular trait may thus not apply to, at least some, wild long-lived species. Prolonged alteration of the HPA axis functioning during ontogeny may not be viable in these species (**Boonstra and Fox, 2013**; **Dantzer et al., 2016**; **Beehner and Bergman, 2017**), and/or, as predicted by the ACM, improvement in the social environment, such as access to buffering mechanisms, such as alloparental care or adoption, allows the HPA axis activity to return to a regular level.

While modifications of HPA axis activity are unlikely to directly impact chimpanzee fitness in adult individuals, the modification in the HPA axis activity during the two first years following maternal loss found in our study might mediate fitness-related costs such as slowed growth (**Samuni et al., 2020**) rather than directly resulting in negative fitness outcomes. In general, future studies on wild mammals should link the effects of early-life adversity on an individual's physiology to long-term fitness consequences in order to gain a clearer understanding of the selective forces at play. An investigation of

the physiological and social differences, between orphans who do survive and those who do not reach maturity, as well as identifying and quantifying the effects of the buffering mechanisms that contribute to these differences, will be key in this process.

## Materials and methods

### Ethics statement

Our study was purely observational and non-invasive. Observers followed the strict hygiene protocol of Taï Chimpanzee Project, which was adopted by IUCN as the best practice guideline for wild ape studies (*Gilardi et al., 2015*; Appendix 1). Observers quarantined for 5 days before following the chimpanzees. During follows, observers disinfected their hands and boots and changed clothes before leaving and entering camps. In the forest, observers wore face masks and kept a minimum distance of 8 m between themselves and the chimpanzees to avoid disease transmission from humans to chimpanzees, and to avoid disturbing the natural behavior of the observed individuals. The research presented here was approved by the 'Ethikrat' of the Max Planck Society on 04.08.2014.

### Study communities

We used the long-term data of the Taï Chimpanzee Project (*Wittig and Boesch, 2019b*) collected on four communities of wild Western chimpanzees (East, North, Middle, and South) in the Taï National Park, Cote d'Ivoire (5°52′N, 7°20′E). The behavioral observation of the chimpanzees started in 1982 and is still ongoing. The observation periods for each of the communities are as follows: North 1982–present; South, 1993–present; Middle, 1995–2004; East, 2000–present (*Wittig, 2018*). Urine samples were collected regularly in all communities from 2000 onwards, except for the East community where sample collection started in 2003.

### Study subjects

For this study, we considered all immature individuals from both sexes (<12 years of age) and mature males (≥12 years of age) from whom urine samples were collected. The age ranges for immatures sampled in this study were 2.82–11.99 years for non-orphans and 4.10–11.99 years for orphans. Physical maturity may come later in male Taï chimpanzees, but 12 years is the age at which chimpanzees range predominantly independently of their mother (Taï Chimpanzee Project, unpublished data) and are fully integrated in the male hierarchy (*Mielke et al., 2018*). We excluded mature females from the analysis since most females in our study immigrated from unhabituated communities into the study communities, which meant we had no knowledge of the presence or absence of their mothers during their immature years. We excluded orphans for whom the date of death of the mother was unknown (e.g., occurred before habituation of the study community). For individual samples, we excluded outliers (i.e., samples with very low or very high hormonal measures, see details in Appendix 1) and samples collected when the individuals were sick. We also ensured that the final dataset comprised at least three data points per individual per year, and that the earliest and the latest samples were separated by at least 6 hr, to ensure a meaningful evaluation of the diurnal cortisol slope of each individual (see details in Appendix 1). In total, we used 846 samples from 50 immatures, including 17 orphans (N samples per individual mean ± SE = 16.9 ± 2.2) and 2184 samples from 28 mature males, including 11 orphans (N samples per individual mean ± SE = 78 ± 13.5). For immatures, we used an average of nine samples per individual per year, and half of the individual_year comprised at least seven samples. For adult males, we used an average of 19 samples per individual per year, and half of the individual_year comprised at least 10 samples.

### Demographic and behavioral data collection

In Taï, each chimpanzee community is followed daily by a joint effort of local and international assistants and researchers (*Wittig and Boesch, 2019b*). Each day, the observers conducted focal follows from and to sleeping sites. The focal individual was either followed all day (i.e., 12 hr) or the identity of the focal changed around 12h30 and two different individuals were followed each day one after the other (i.e., 6 hr focal follow each). The observer recorded detailed focal and ad libitum behavioral data (*Altmann, 1974*). Observers recorded all social interactions such as aggression and submissive behaviors, which we then used to build a dominance hierarchy (see below). In addition, each day, the

observers recorded the presence of all individual chimpanzees they encountered, which provides a detailed account of the demography of each community. Specifically, we obtained detailed information on individuals' date of birth, immigration/emigration, and death or disappearance. This information was used to determine the early-life history of the study subject, namely if their mother died before they reached 12 years of age, and, if so, the age of the subject when its mother died.

## Assessment of dominance hierarchy in mature males

Since dominance rank may correlate with the cortisol levels of adult male chimpanzees (e.g., *Muller and Wrangham, 2004*), we wanted to control for this parameter in our analysis of cortisol patterns in adult males. We calculated the dominance hierarchy for mature males in each of the study communities using a modified version of the Elo-rating method (*Neumann et al., 2011*) developed by *Foerster et al., 2016*. In this modified version, the *k* parameters and the starting score of each individual are optimized using maximum likelihood approximation (*Mielke et al., 2018*, see details in Appendix 1). We used all of the long-term data available on unidirectional submissive pant-grunt vocalizations, given by the lower ranking of the two individuals towards the higher ranking (*Bygott, 1979*). We used 9189 pant-grunt recorded for males in Taï South, 3952 in Taï East, 5784 in Taï North, and 111 in Taï Middle. All Elo-rating scores were standardized between 0 and 1 with 1 being the highest-ranking individual and 0 the lowest ranking on any given day. We then extracted the Elo-rating score of each individual on the day when each urine sample was collected.

## Urine sample collection and analysis

During chimpanzee follows, we collected urine samples opportunistically from known individuals. Directly after urination, we collected the urine from leaves and/or the ground into a 2 ml cryo vial using a disposable plastic pipette. Within 12 hr of collection, we placed these vials in liquid nitrogen. Subsequently, the samples were shipped on dry ice to the Endocrinology Laboratory of the Max Plank Institute for Evolutionary Anthropology in Leipzig, Germany, and stored at ≤–20°C until analysis. We usedLCMS ( *Hauser et al., 2008*; *Murtagh et al., 2013*) and MassLynx (version 4.1; QuanLynx-Software) to quantify cortisol concentrations in each sample. For all samples analyzed, we used either prednisolone (hereafter the 'old' method; i.e., most samples analyzed prior to July 2016) or cortisol d4 (hereafter the 'new' method; i.e., all samples analyzed after September 2016) as our internal standard for the cortisol measurements. To adjust for water content in the urine (i.e., urine concentration), we measured, for each sample, its specific gravity (SG) using a refractometer (TEC, Ober-Ramstadt, Germany). We corrected our cortisol concentration for urine water content in each sample using the following formula provided by *Miller et al., 2004*:

$$\text{SG corrected cortisol} = \text{raw cortisol concentration (ng/ml)} \times \frac{(\text{SG}_{\text{population mean}} - 1.0)}{(\text{SG}_{\text{sample}} - 1.0)}$$

where $\text{SG}_{\text{population mean}}$ is the mean SG value average across all the samples used in this study ($\text{SG}_{\text{population mean}}$ = 1.022 in our study), and $\text{SG}_{\text{sample}}$ is the SG value of each given sample. In the article, all cortisol concentrations are reported as ng/ml SG.

## Statistical analysis

We used a series of Bayesian LMMs to test our predictions regarding the effect of maternal loss on overall cortisol levels and diurnal slopes (jointly constituting the cortisol profile). We first tested these effects in socially immatures from both sexes (*all immature* and *immature orphan models*). Secondly, we tested these effects in mature males in the *all adult male* and *adult male orphan model*s (i.e., four models in total). In all the models, each urine sample represented a data point and the cortisol concentration of the sample (expressed in ng/ml SG) was the response variable. We log-transformed the cortisol values to achieve a symmetric distribution of the response.

## Effect of maternal loss on cortisol profiles in immatures

In the *all immature model*, we tested the prediction that immature orphans, especially those recently orphaned, will have higher overall cortisol levels and a flatter slope of diurnal cortisol slope when compared to non-orphans. We fitted a LMM with a categorical variable for orphan status with three levels (recently orphaned, non-recently orphaned, and non-orphans) as our test predictor to test for

the short- and long-term effects of maternal loss on overall cortisol levels in immatures. In addition, to test for the effect of this orphan status on the diurnal cortisol variation, we incorporated the linear, quadratic, and cubic terms for *time of sample collection* as test predictors as well as their interaction with orphan status to test whether the diurnal cortisol slope differed between recently orphaned, non-recently orphaned, and non-orphans. The *time of sample collection* was expressed in minutes with 0 being midnight and 720 being noon. In addition, in this model, we used the following control predictors: the LCMS method used ('old' or 'new' method, see the Urine analysis section), sex of the individual, community size, sex ratio of mature individuals in the community and age of the individual, since these factors can all influence the cortisol profiles of immature and/or mature chimpanzees (*Muller and Wrangham, 2004*; *Emery Thompson et al., 2020*; *Tkaczynski et al., 2020*). Community ID was not included as a control predictor in our analysis since it was highly correlated with community size. In addition, we accounted for seasonal variation in ecological conditions (e.g. rainfall, temperature, food availability) that can affect cortisol levels in chimpanzees (*Wessling et al., 2018*; *Preis et al., 2019*; *Samuni et al., 2019b*) by converting the Julian date at which samples were collected into a circular variable and including the sine and cosine of this variable in our model (*Wessling et al., 2018*).

## Effect of age at mother's death and time since mother died on cortisol profile in orphan immatures

In the *immature orphan model*, we focused on immatures who were orphans at the time of sampling in order to investigate more specifically the effects of the age at maternal loss and years since maternal loss on cortisol levels. We tested the predictions that (a) immature individuals who were orphaned at a younger age would have higher cortisol levels and flatter diurnal cortisol slopes and (b) that, if some form of recovery occurs, these effects will be weaker the more time has passed since an individual lost its mother. We incorporated two test predictors in the *immature orphan model*, the age at which the individuals have been orphaned (in days since their date of birth), and the time since the individuals have been orphaned (in days since the date their mother died). As in the *all immature model*, we incorporated the six interaction terms between these two test predictors and the linear, quadratic, and cubic terms for *time of sample collection*. As before, we also incorporated individual sex, community size, sex ratio, and LCMS method, and the sine and cosine of the Julian date as control fixed effect. Initially, we also wanted to incorporate the age of the individual at the time of sampling into the *immature orphan model*. This was however not possible due to collinearity between the individual age at sample and both the age at which the individual was orphaned and the time since its mother died (i.e., the model did not run due to collinearity issues). However, since age at sample did not have a consistent effect in our immature sample in the *all immature model* (90% CI: –0.27: 0.29) we decided to not include age at sample in the *immature orphan model*, allowing us to test our variable of interest, namely years since maternal loss and age when mother died.

## Effect of maternal loss on cortisol profiles in mature males

In the *all adult male model*, we tested whether mature males (≥12 years) who were orphaned as immatures (i.e., before 12 years of age) had overall higher cortisol levels and a flatter diurnal cortisol slope than mature males who did not lose their mother before 12 years of age. We fitted a LMM with the early-life orphan status (i.e., 'no', if the mother of the individual was still alive when the individual reached 12 years of age, and 'yes' if the mother died before the individual was 12 years of age) as our test predictor. As in the *all immature model*, we incorporated three interaction terms between orphan status and the linear, quadratic, and cubic terms for *time of sample collection*. As in the other models, we used community size, the sex ratio, the age of the individual, the LCMS method, and the sine and cosine of the Julian date as control fixed effects. In addition, we controlled for the dominance rank of the individual by adding the standardized Elo-rating score of each individual on the day the sample was collected as a fixed effect into the model.

## Effect of age at mother's death on cortisol profile in orphan immatures

In the *adult male orphan model*, we assessed whether the age at which the orphan adult male chimpanzees lost their mother impacted the diurnal cortisol levels and slopes of mature males (i.e., whether the potential effect of early-life adversity continued into adulthood). Accordingly, we fitted a

LMM with 'age at which mother died' as a test predictor and its interaction with the linear, quadratic, and cubic terms for time of day. In this model, we used only samples collected from mature males who lost their mothers before they were 12 years of age. As previously, we used community size, the sex ratio, the age of the individual, the LCMS method, and the sine and cosine of the Julian date as control fixed effects.

In addition to the fixed effects, in all of the LMMs we included individual identity as a random factor to avoid pseudoreplication. To control for the changes in cortisol diurnal slope with age, we built one slope per individual per year into each model by incorporating as random factor a dummy variable 'individual_year'. In addition, since certain years might have particularly harsh or favorable ecological conditions, and since this can affect cortisol levels in primates (e.g., *Young et al., 2019*), we also included year as a random factor in each model. Finally, our hormonal dataset included samples collected by different observers with different research interests (hereafter project). Whilst all projects followed a similar design to collect at least one urine sample from focal individuals throughout the day, some projects conducted additional targeted sampling of specific behaviors such as aggressions or affiliations. Thus, to account for potential variation in cortisol levels that may be a result of inter-observer project bias, we added the 'project' type as an additional random factor.

All analyses were conducted in R 4.0.3 (R core Team 2020) using the function *brm* from the package 'brms' (*Bürkner, 2018*). In each model, we included the maximal random slope structure between each fixed predictor (test and control) and each random effect (*Baayen et al., 2008*; *Barr et al., 2013*) and the correlation between intercept and slopes. In particular, the linear, quadratic, and cubic terms for time of day were included as random slopes within each of the random effects. For each model, we extracted both the 95% and 90% CIs for each fixed effect and for each random effect from the posterior distribution. In all the models, we used weakly regularizing priors for the fixed effects (Normal (0,1)) and the priors given by default by the function 'get_prior' of the package 'brms' for the random effects (i.e., Student t (3, 0, 2.5) for the random intercepts and slopes and lkj (1) for their correlation). We chose weakly regularizing prior for the fixed effects since they give less weight to outlier data points and therefore help constrain model predictions to biologically meaningful estimates and CI (*Lemoine, 2019*).

Before fitting each model, we tested for collinearity issues between our predictor variables by computing the variance inflation factor (VIF) using the function *vif* from the package 'car' (*Fox and Weisberg, 2011*). Collinearity was not an issue in any of the final models (VIF of all predictor variables < 3.6). Sampling diagnostics (Rhat <1.1) and trace plots confirmed chain convergence for all models. Effective sample sizes (all >1000) confirmed no issues with autocorrelation of sampling for all models. Please note that the effective sample size is a measure of autocorrelation and does not correspond to the number of data points that were used for each model (namely 393, 846, 2184, and 769 for the *all immature*, the *orphan immature*, the *all adult male*, and the *adult male orphan model*s, respectively). After running the models, we processed to a posterior predictive check using the function 'pp_check' from the package 'brms' (*Appendix 1—figure 2*, *Appendix 1—figure 3*, *Appendix 1—figure 4*, *Appendix 1—figure 5*).

## Acknowledgements

We are very grateful to Christophe Boesch for his years of dedication to building the Taï Chimpanzee Project and amassing impressive long-term data, and for engaging in massive and critical conservation efforts to ensure the ongoing survival of West African Chimpanzees. We thank the Ministère de l'Enseignement Supérieur et de la Recherche Scientifique and the Ministère de Eaux et Fôrets in Côte d'Ivoire, and the Office Ivoirien des Parcs et Réserves for permitting the study. We are grateful to the Centre Suisse de Recherches Scientifiques en Côte d'Ivoire and to Tatiana Bortolato and Lara Southern and to the staff members of the Taï Chimpanzee Project for their support and collecting the data. We thank three anonymous reviewers and the editor for very constructive comments on a previous version of this manuscript. This study was funded by the Max Planck Society and the European Research Council (ERC) under the European Union's Horizon 2020 research and innovation program awarded to CC (grant agreement no. 679787). Core funding for the Taï Chimpanzee Project has been provided by the Max Planck Society since 1997.

## Additional information

### Funding

| Funder | Grant reference number | Author |
| --- | --- | --- |
| H2020 European Research Council | 679787 | Catherine Crockford |

The funders had no role in study design, data collection and interpretation, or the decision to submit the work for publication.

### Author contributions

Cédric Girard-Buttoz, Conceptualization, Data curation, Formal analysis, Investigation, Methodology, Visualization, Writing – original draft; Patrick J Tkaczynski, Conceptualization, Data curation, Methodology, writing-review-and-editing; Liran Samuni, Cristina Gomes, Anna Preis, Data curation, writing-review-and-editing; Pawel Fedurek, writing-review-and-editing; Therese Löhrich, Virgile Manin, Prince F Valé, Data curation; Tobias Deschner, Data curation, Methodology, resources, writing-review-and-editing; Roman M Wittig, Conceptualization, Data curation, funding-acquisition, Methodology, project-administration, resources, supervision, writing-review-and-editing; Catherine Crockford, Conceptualization, Data curation, funding-acquisition, Investigation, Methodology, project-administration, supervision, writing-review-and-editing

### Author ORCIDs

Cédric Girard-Buttoz http://orcid.org/0000-0003-1742-4400
Liran Samuni http://orcid.org/0000-0001-7957-6050
Roman M Wittig http://orcid.org/0000-0001-6490-4031
Catherine Crockford http://orcid.org/0000-0001-6597-5106

### Ethics

The research presented here was non-invasive and did not comprise experimental work. Our work comprised only behavioral observations from a distance and non-invasive collection of urine samples. This work was approved by the 'Ethikrat' of the Max Planck Society and the European Research Council ethics board under grant agreement no. 679787.

### Decision letter and Author response

Decision letter https://doi.org/10.7554/eLife.64134.sa1
Author response https://doi.org/10.7554/eLife.64134.sa2

## Additional files

### Supplementary files

• Transparent reporting form

### Data availability

The data have been deposited on Dryad: https://doi.org/10.5061/dryad.gtht76hk2.

The following dataset was generated:

| Author(s) | Year | Dataset title | Dataset URL | Database and Identifier |
| --- | --- | --- | --- | --- |
| Girard-Buttoz C, Tkaczynski PJ, Samuni L, Fedurek P, Gomes C, Löhrich T, Manin V, Preis A, Valé PF, Deschner T, Wittig RM, Crockford C | 2020 | | https://doi.org/10.5061/dryad.gtht76hk2 | Dryad Digital Repository, 10.5061/dryad.gtht76hk2 |

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

# Appendix 1

## Data preparation

The initial dataset comprised 4604 samples (1518 from immature individuals and 3086 from mature males) from 46 female and 48 male immature individuals and 34 mature males. We applied a suite of selection criteria to subset our dataset to samples collected from individuals from whom all demographic and social data needed were available. We excluded all individuals for whom we could not assess if the mother died before they were 12 years of age or the age they were when their mother died. We also excluded samples for whom the cortisol concentration could not be measured or was excessively low (<0.1 ng/ml SG). We excluded samples with very low SG (SG <1.003). Very low SG values are a sign of over-diluted samples that reflect potential contamination with rain water and can, in turn, inflate cortisol concentration measurements. We also excluded samples collected from individuals on days when they displayed injuries or symptoms of sickness (as assessed by the on-site veterinary staff) since injury and sickness lead to extremely elevated cortisol levels in primates (e.g., Barton 1987; Muehlenbein & Watts 2010; *Behringer et al., 2020*). Finally, since a large part of our analysis focused on circadian cortisol variation, we excluded all samples for which we did not have a precise time of collection recorded. For the same reason, we limited our dataset for each individual to years when at least three samples were collected from this specific individual, and in years in which the earliest and the latest sample collections were separated in time by at least 6 hr. This criterion was applied in order to be able to calculate, in our statistical model, a meaningful circadian slope for each individual each year with time variation representing at least half of the active time of the chimpanzees (i.e., at least 6 hr out of 12 hr). The three samples could have been collected on different days, but 'time of sample collection' was used to define the 6 hr criteria. Following this selection process, we were left with 849 samples from 50 immatures (including 17 orphans) and 2184 samples from 28 mature males (including 11 orphans).

**Appendix 1—table 1.** Results of the all adult male model testing the effect of maternal loss on cortisol profiles in all mature males.

'All adult male model full' refers to the full model ran with all the interactions considered. 'All adult male model reduced' refers to the reduced model after removing the interactions for which the 90% credible interval (CI) overlapped 0. SE indicates the standard error of the estimate for each predictor. The coded level for each categorical predictor is indicated in parentheses. Control predictors are italicized. 95% CI low and 95% CI high indicate the lower and upper limits of the 95% CI. Likewise, 90% CI low and 90% CI high indicate the lower and upper limits of the 90% CI. CIs that do not overlap 0 are indicated in bold. LCMS: liquid chromatography mass spectrometry. Time of the day$^2$: quadratic term for time of the day. Time of the day$^3$: cubic term for time of the day.

| Model | Response | Predictor | Estimate | SE | 95% CI low | 95% CI high | 90% CI low | 90% CI high |
|---|---|---|---|---|---|---|---|---|
| | | Intercept | 3.97 | 0.25 | 3.44 | 4.44 | 3.55 | 4.36 |
| | | Time of the day | –0.48 | 0.09 | **–0.67** | **–0.32** | **–0.63** | **–0.35** |
| | | Time of the day$^2$ | –0.07 | 0.05 | –0.17 | 0.03 | –0.15 | 0.01 |
| | | Time of the day$^3$ | 0.03 | 0.04 | –0.05 | 0.12 | –0.04 | 0.1 |
| | | Orphan status (yes, orphan) | 0.02 | 0.19 | –0.36 | 0.38 | –0.29 | 0.32 |
| | | Individual age at sample | 0.19 | 0.14 | –0.07 | 0.49 | –0.03 | 0.42 |
| | | LCMS method (old) | 0.17 | 0.31 | –0.41 | 0.82 | –0.31 | 0.69 |
| | | Community size | 0.06 | 0.09 | –0.13 | 0.24 | –0.1 | 0.21 |
| All adult male model full | Log urinary cortisol levels (ng/ml SG) | Sex ratio | 0.04 | 0.07 | –0.11 | 0.18 | –0.09 | 0.16 |
| | | Dominance rank | 0.04 | 0.08 | –0.12 | 0.19 | –0.09 | 0.17 |

*Appendix 1—table 1 Continued on next page*

*Appendix 1—table 1 Continued*

| Model | Response | Predictor | Estimate | SE | 95% CI low | 95% CI high | 90% CI low | 90% CI high |
|---|---|---|---|---|---|---|---|---|
| | | Sin(seasonDate) | –0.04 | 0.02 | **–0.08** | **0** | **–0.07** | **0** |
| | | Cos(seasonDate) | –0.01 | 0.02 | –0.05 | 0.03 | –0.04 | 0.03 |
| | | Orphan status (yes): time of the day | 0.02 | 0.04 | –0.05 | 0.1 | –0.04 | 0.09 |
| | | Orphan status (yes): time of the day$^2$ | 0.01 | 0.04 | –0.08 | 0.09 | –0.06 | 0.08 |
| | | Orphan status (yes): time of the day$^3$ | –0.03 | 0.08 | –0.18 | 0.13 | –0.15 | 0.1 |
| All adult male model reduced | Log urinary cortisol levels (ng/ml SG) | Intercept | 3.97 | 0.24 | 3.45 | 4.43 | 3.55 | 4.35 |
| | | Time of the day | –0.49 | 0.08 | **–0.67** | **–0.33** | **–0.64** | **–0.36** |
| | | Time of the day$^2$ | –0.06 | 0.04 | –0.15 | 0.03 | –0.14 | 0.01 |
| | | Time of the day$^3$ | 0.04 | 0.04 | –0.03 | 0.12 | –0.02 | 0.11 |
| | | Orphan status (yes, orphan) | 0.03 | 0.18 | –0.33 | 0.38 | –0.28 | 0.32 |
| | | Individual age at sample | 0.19 | 0.14 | –0.07 | 0.48 | –0.02 | 0.42 |
| | | LCMS method (old) | 0.17 | 0.3 | –0.41 | 0.8 | –0.3 | 0.68 |
| | | Community size | 0.05 | 0.09 | –0.13 | 0.24 | –0.1 | 0.21 |
| | | Sex ratio | 0.03 | 0.07 | –0.11 | 0.17 | –0.09 | 0.15 |
| | | Dominance rank | 0.04 | 0.08 | –0.12 | 0.19 | –0.09 | 0.17 |
| | | Sin(seasonDate) | –0.04 | 0.02 | **–0.08** | **0** | **–0.07** | **0** |
| | | Cos(seasonDate) | –0.01 | 0.02 | –0.05 | 0.03 | –0.04 | 0.03 |

**Appendix 1—table 2.** Results of the adult male orphan model testing, in orphan mature males only, the effect of age at maternal loss on cortisol profiles.

'Adult male orphan model full' refers to the full model ran with all the interactions considered. 'Adult male orphan model reduced' refers to the reduced model after removing the interactions for which the 90% credible interval (CI) overlapped 0.SE indicates the standard error of the estimate for each predictor. The coded level for each categorical predictor is indicated in parentheses. Control predictors are italicized. 95% CI low and 95% CI high indicate the lower and upper limits of the 95% CI. Likewise, 90% CI low and 90% CI high indicate the lower and upper limits of the 90% CI. CIs that do not overlap 0 are indicated in bold. LCMS: liquid chromatography mass spectrometry. Time of the day$^2$: quadratic term for time of the day. Time of the day$^3$: cubic term for time of the day.

| Model | Response | Predictor | Estimate | SE | 95% CI low | 95% CI high | 90% CI low | 90% CI high |
|---|---|---|---|---|---|---|---|---|
| | | Intercept | 4.07 | 0.33 | 3.41 | 4.71 | 3.53 | 4.58 |
| | | Time of the day | –0.5 | 0.13 | –0.77 | –0.24 | –0.72 | –0.29 |
| | | Time of the day$^2$ | –0.08 | 0.07 | –0.21 | 0.05 | –0.19 | 0.03 |
| | | Time of the day$^3$ | 0.04 | 0.06 | –0.09 | 0.17 | –0.06 | 0.14 |
| | | Orphan's age when mother died | –0.12 | 0.25 | –0.62 | 0.37 | –0.53 | 0.29 |
| | | Community size | 0.04 | 0.21 | –0.34 | 0.48 | –0.27 | 0.4 |
| | | Sex ratio | 0.02 | 0.17 | –0.31 | 0.34 | –0.25 | 0.29 |
| | | Dominance rank | –0.07 | 0.18 | –0.41 | 0.3 | –0.36 | 0.24 |
| | | Orphan's age at sample | 0.43 | 0.37 | –0.23 | 1.24 | –0.12 | 1.08 |
| Adult male orphan full | Log urinary cortisol levels (ng/ml SG) | LCMS method (old) | 0.03 | 0.31 | –0.55 | 0.66 | –0.45 | 0.54 |

*Appendix 1—table 2 Continued on next page*

*Appendix 1—table 2 Continued*

| Model | Response | Predictor | Estimate | SE | 95% CI low | 95% CI high | 90% CI low | 90% CI high |
|---|---|---|---|---|---|---|---|---|
| | | Sin(seasonDate) | –0.01 | 0.03 | –0.08 | 0.05 | –0.07 | 0.04 |
| | | Cos(seasonDate) | –0.01 | 0.03 | –0.08 | 0.06 | –0.07 | 0.05 |
| | | Orphan's age when mother died: time of the day | 0.03 | 0.07 | –0.11 | 0.18 | –0.09 | 0.04 |
| | | Orphan's age when mother died: time of the day$^2$ | 0.06 | 0.04 | –0.02 | 0.14 | –0.01 | 0.12 |
| | | Orphan's age when mother died: time of the day$^3$ | –0.02 | 0.04 | –0.11 | 0.06 | –0.09 | 0.15 |
| Adult male orphan reduced | Log urinary cortisol levels (ng/ml SG) | Time of the day | 4.07 | 0.33 | 3.43 | 4.72 | 3.55 | 4.6 |
| | | Time of the day$^2$ | –0.51 | 0.13 | –0.77 | –0.26 | –0.72 | –0.3 |
| | | Time of the day$^3$ | –0.07 | 0.07 | –0.21 | 0.06 | –0.19 | 0.04 |
| | | Orphan's age when mother died | 0.04 | 0.06 | –0.08 | 0.17 | –0.05 | 0.14 |
| | | Community size | –0.08 | 0.25 | –0.59 | 0.41 | –0.5 | 0.33 |
| | | Sex ratio | 0.04 | 0.21 | –0.36 | 0.49 | –0.28 | 0.39 |
| | | Dominance rank | 0.02 | 0.17 | –0.33 | 0.33 | –0.27 | 0.28 |
| | | Orphan's age at sample | –0.07 | 0.18 | –0.42 | 0.31 | –0.36 | 0.24 |
| | | LCMS method (old) | 0.43 | 0.38 | –0.24 | 1.26 | –0.12 | 1.09 |
| | | Sin(seasonDate) | 0.02 | 0.31 | –0.58 | 0.66 | –0.47 | 0.54 |
| | | Cos(seasonDate) | –0.01 | 0.03 | –0.08 | 0.06 | –0.07 | 0.05 |
| | | Time of the day | –0.01 | 0.03 | –0.08 | 0.06 | –0.07 | 0.04 |

**Appendix 1—table 3.** List of orphan immatures in the study and information about the adoption by adult individuals in the community.

| Identity | Sex | Community | Age when mother died (years) | Adopted?* | Duration of adoption* | No. of samples in the study | Age when sampled for the study |
|---|---|---|---|---|---|---|---|
| Beatrice | F | East | 4.9 | Yes | >3 years | 38 | 5.3–8.6 |
| Emma | F | East | 4.1 | Yes | >2 years | 17 | 4.1–6.1 |
| Eolos | M | East | 3.9 | Yes | >3 years | 7 | 7.0–7.3 |
| Erasmus | M | East | 8.7 | Unknown | | 31 | 8.8–11.8 |
| Fatima | F | East | 6.6 | Unknown | | 8 | 7.8–10.1 |
| Gia | F | East | 2.6 | Yes | 17 months | 8 | 10.6–11.9 |
| Maimouna | F | East | 4.7 | No | | 20 | 6.7–8.9 |
| Quarantine | F | East | 5.5 | Yes | >2 years | 4 | 7.6–7.9 |
| Richelieu | M | East | 5.4 | Unknown | | 64 | 10.7–11.8 |
| Willy | M | East | 10.5 | Unknown | | 57 | 10.5–11.9 |
| Baloo | F | South | 3.8 | yes | 1 year | 22 | 7.3–8.5 |
| Caramel | M | South | 7.3 | Unknown | | 4 | 8.7 – 9.3 |
| Mohan | F | South | 4.1 | Yes | 1.5 year | 34 | 4.1 – 6.5 |
| Oscar | M | South | 4.7 | Yes | >2 years | 49 | 8.1 – 11.9 |
| Wala | F | South | 4.9 | Unknown | | 25 | 8.1 – 10.9 |
| Roxane | F | North | 4.7 | Unknown | | 4 | 10.5 – 11.3 |
| Volta | F | North | 3.8 | Unknown | | 3 | 10.7 – 10.7 |

*Data taken from **Samuni et al., 2019a**.

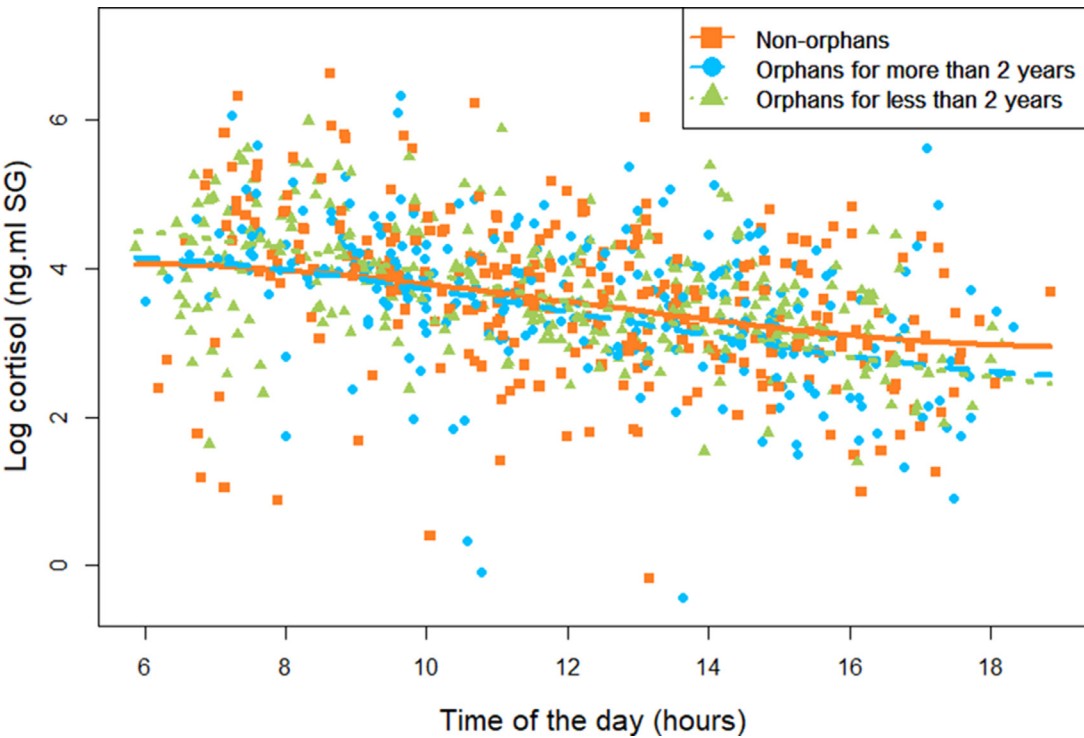

**Appendix 1—figure 1.** Diurnal cortisol level variation in immature chimpanzees. Non-orphans are depicted by orange squares, recently orphans (orphaned for less than 2 years) by green triangles, and non-recently orphaned (orphaned for more than 2 years) in blue circles. Each dot represents one sample. As for *Figure 1*, the orange solid line and green and blue dashed lines depict the model prediction lines from the all immature model Model 1a for non-orphans, recently orphans, and non-recently orphans, respectively. The model lines depict the consistent effect of the interaction between orphan status and the linear term for time of day in the all immature model Model 1a (estimate: –0.22, 95% credible interval [CI]: [–0.03: –0.48]). The sample size for the all immature model Model 1a was N = 846 samples and 50 individuals.

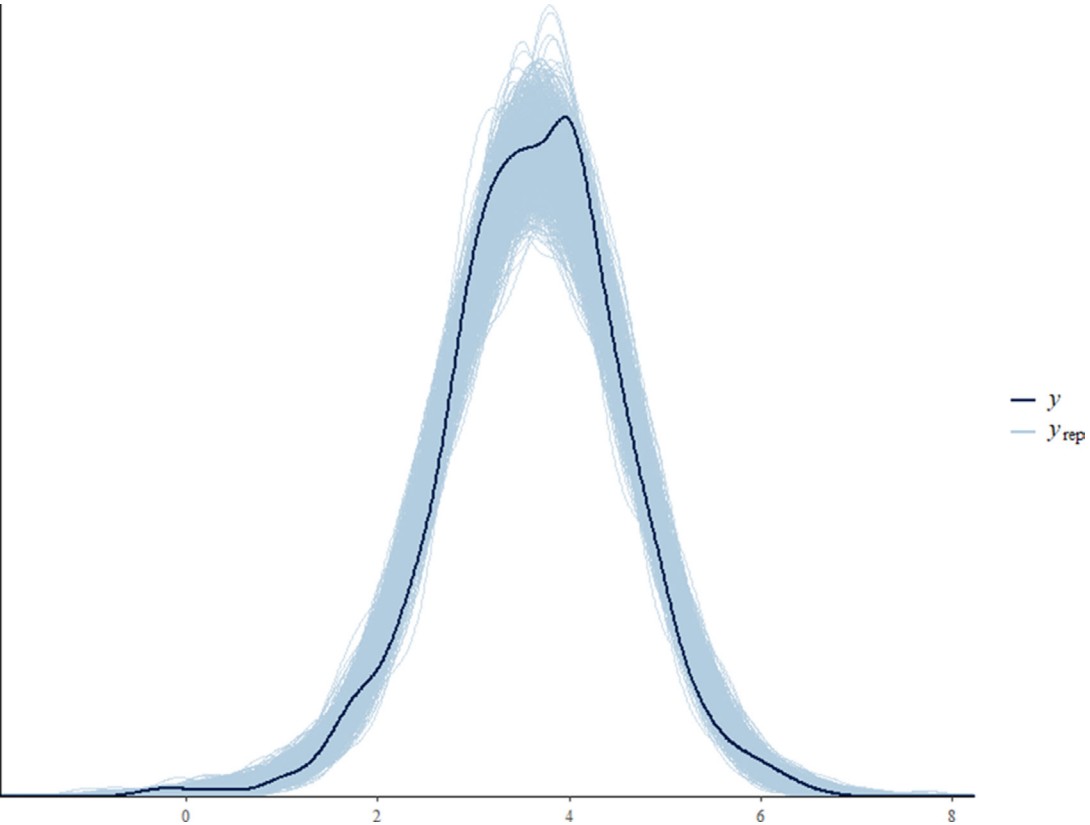

**Appendix 1—figure 2.** Posterior predictive check for the all immature model.

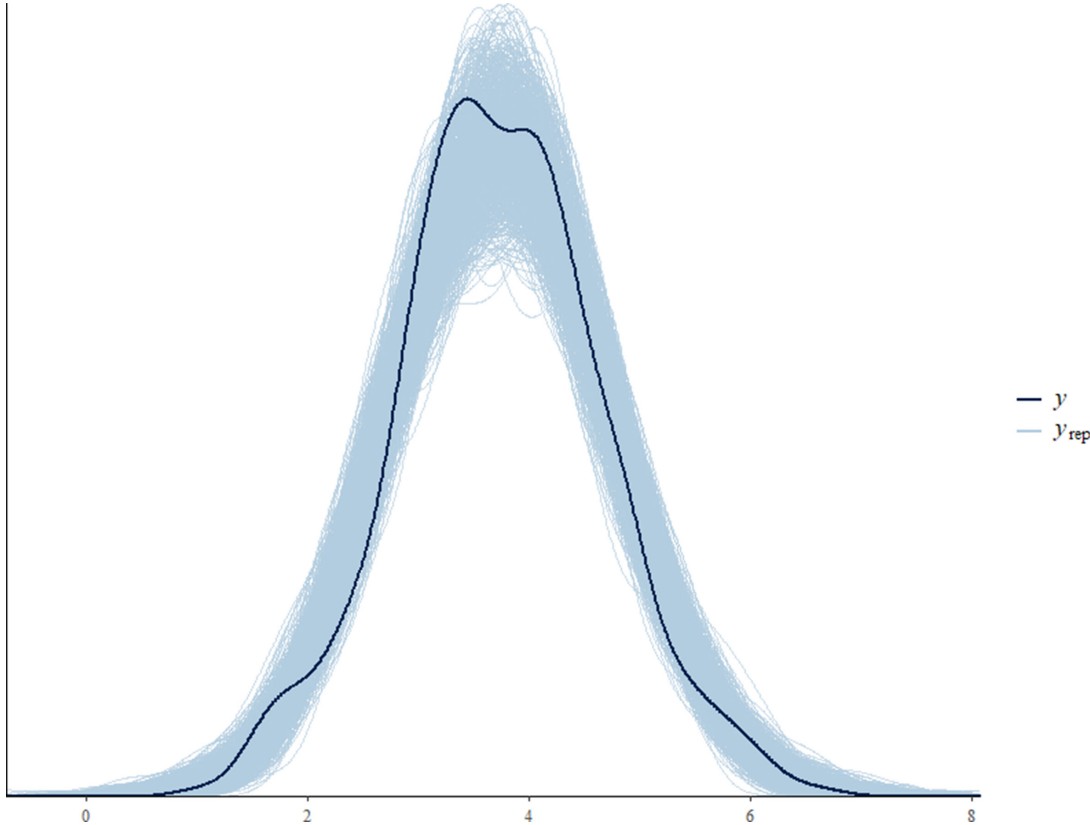

**Appendix 1—figure 3.** Posterior predictive check for the immature orphan model.

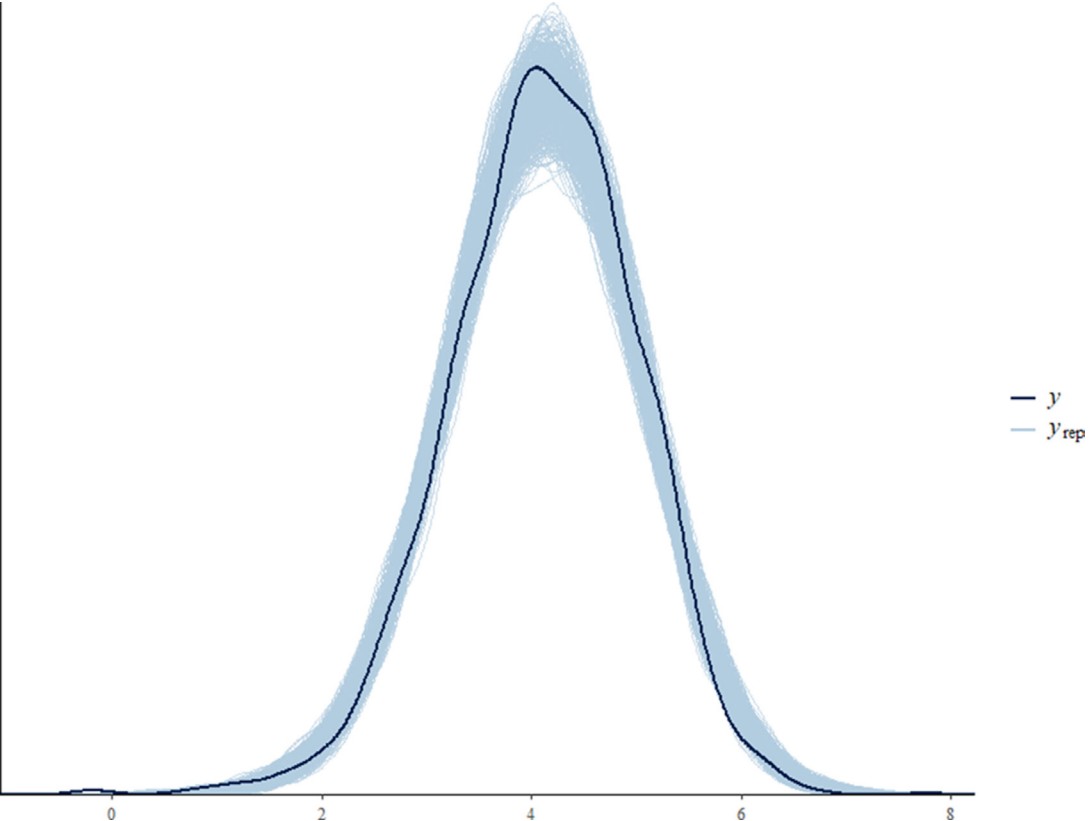

**Appendix 1—figure 4.** Posterior predictive check for the all adult male model.

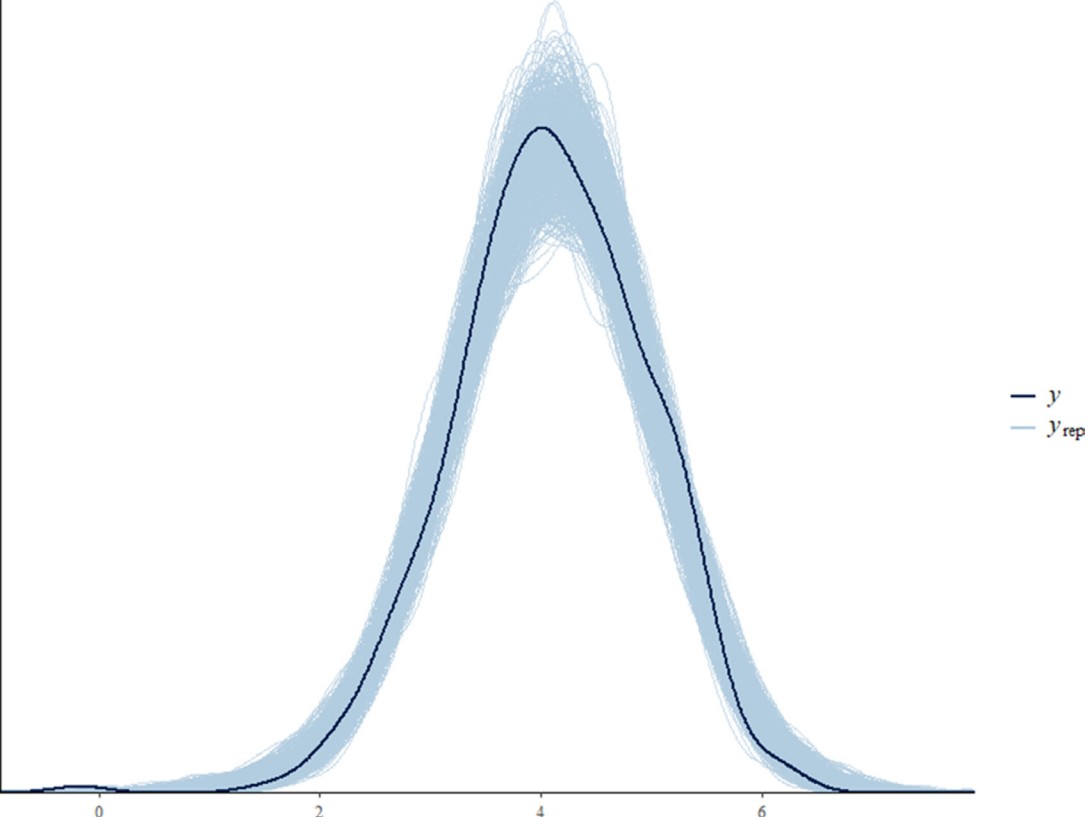

**Appendix 1—figure 5.** Posterior predictive check for the adult male orphan model.

