## [Decision Letter]

**Acceptance summary:**

This study examined the relationship between maternal loss and cortisol excretion – a proxy for stress, among wild chimpanzees in the Tai Forest of Cote d'Ivoire. It tests whether and how the effect of early maternal loss is reflected in individual cortisol levels. The authors found that diurnal cortisol slopes across the day differed between immature chimps who lost their mothers early and those who did not, but that this difference is not visible later in life, suggesting a short-term alteration of the hypothalamic-pituitary-adrenal axis – a finding consistent with the adaptive calibration model rather than the biological embedding model that proposes a long-term alteration of the hypothalamic-adrenal-axis due to early maternal loss. The study is one of the rare empirical tests of the biological embedding and the adaptive calibration models on wild long-lived mammals and opens opportunities for further investigation of the impact of early life experience on the resilience of non-human primates which are especially threatened by hunting and habitat destruction. Future studies should, however, address the possibility that the lack of a later life association between maternal loss and cortisol levels may be due to selective early mortality of individuals with high cortisol levels. Social and environmental factors that may buffer the effect of early maternal loss should also be considered.

**Decision letter after peer review:**

Thank you for submitting your article "Early maternal loss affects diurnal cortisol slopes in immature but not mature wild chimpanzees" for consideration by *eLife*. Your article has been reviewed by 3 peer reviewers, and the evaluation has been overseen by a Reviewing Editor and George Perry as the Senior Editor. The reviewers have opted to remain anonymous.

The reviewers have discussed the reviews with one another and the Reviewing Editor has drafted this decision to help you prepare a revised submission.

Summary:

This paper tests the biological embedding model by asking whether and how early maternal loss effects cortisol levels and diurnal cortisol slopes among wild chimpanzees at the Tai Forest, Cote d'Ivoire. The results suggest that maternal loss alters the HPA stress axis in wild chimpanzees, but these effects are not visible later in life. Authors suggest that the lack of a later life association between maternal loss and cortisol levels may be due to selective early mortality of individuals with high cortisol levels but did not provide any survival or behavioural data to show that orphans and non-orphans differ in any fitness-related traits other than cortisol. Furthermore, the association between cortisol and the HPA axis is in the opposite direction to that observed in humans and there seems to be no significant increase in cortisol in orphans compared to non-orphans. Overall, the study is the result of extensive fieldwork and the number of samples collected is impressive. The subject is very interesting, and we generally agree that with an extensive reworking of the entire framework and analyses, it could be a good fit for *eLife*.

The analyses will benefit greatly if the authors use effect sizes and confidence intervals for inferences instead of p-values. This may solve the significance threshold issues. Moreover, the reliance on p-values seem to limit the value of the data. For example, authors suggest that results from model 1 should be treated with caution because the full model is not significantly different from the null model, but by relying on it as the key finding of the study without exploring effect sizes, it does not seem that they did exercise sufficient caution.

Please find more specific comments below:

Essential revisions:

1. Present results as effect sizes with confidence intervals and make inferences along the line of the percentage (or ng/ml) by which orphans differ from non-orphans and over time. This effect sizes can be more easily compared with results from human studies on cortisol. Please communicate findings more clearly and discuss exactly why the pattern in this Chimp population may be different from that in humans. Pay attention to the following comment from reviewers:

a. Despite acknowledging that the "significance of these predictors should be interpreted with caution" because model 1a did not reach significance, the authors make very strong claims about the results in the discussion- and also feature the finding of that model in the title of the paper. That seems problematic to me- especially because the insignificant model results (more intense diurnal slopes among immature orphans) diverge from the expectations set forth by other works in humans and non-humans. The finding that this is to do with higher-than-expected morning cortisol is puzzling given that evening levels are generally considered more responsive or plastic. However, this could also be an artefact of fitting the models without the third-order term for time.

2. The lack of significance could be due to insufficient sampling or a true lack of predictive power. Reviewers provide specific suggestions on how to reanalyse the data given the difficulty of collecting additional samples currently.

a. Model 1B and figure 2 demonstrate that the cortisol response to maternal loss declines over time and that after 2 years it is no longer detectably different from non-orphans. The authors do not account for age since maternal loss in model 1A. If a considerable proportion of samples were from orphans that lost their mothers more than 2 years ago, this would reduce the likelihood of detecting a significant difference between orphans and non-orphans and potentially explain the lack of significance in the overall model. Crucially I think if model 1a was adjusted to separate out recent orphans from those that lost their mothers less recently this could enable the authors to better back up their claims at least in relation to changes in overall cortisol levels.

b. The truth is that cortisol data are very messy and even though 300+ samples from 50 individuals might seem like a lot, it might turn out that it isn't enough to detect a signal. At other sites, cortisol levels and diurnal slopes shift with age- and this is true for humans as well. However, the slope should be more susceptible than more average levels so the authors might be able to make stronger conclusions based on average or time-corrected cortisol rather than focusing so much on a slope. Either way- though improved modelling to access slope or by setting slope aside and focusing on average cortisol- the data here certainly have a path to publication

c. My principal concerns with this paper, as written, revolve around the methods/results. First and foremost, I am not convinced that the authors have a sufficient sample size to evaluate the predictions/hypotheses outlined in the introduction. While 849 urine samples are a large number, and again, their efforts here should be commended, the sample spread is quite thin once it is spliced up into appropriate categories, especially considering how many samples were collected per individual year, on average. As the authors indicate throughout and especially when describing their modelling approach, cortisol is inherently a very noisy hormone impacted by a myriad of factors- including age in at least one other densely sampled chimpanzee community. I am also surprised that time of day was modelled quadratically. It is my understanding that humans, other populations of chimpanzees, and other mammals follow a sigmoidal curve which should be modelled with a third-order term as well. For these reasons, it is difficult to tell whether model 1A is not significant because of the insufficient sample or a true lack of predictive power. Additionally, I am concerned that the paper seems to focus so much on the results from a single model term in a model that did not reach significance.

d. It would be useful to see some of the raw data- especially a plot showing cortisol values across the time of day. Regarding that third-order term- it isn't out of the question that T + T^2 would be sufficient, however, I do believe it's important to rule out that including T^3 does a better job.

e. L449 As slopes are calculated for each individual each year, what is the mean number of samples per individual per year? 3 minimum seems very small for calculating a slope but if the average is considerably higher, then perhaps it is not an issue.

f. Please include more information about model results, sample size, etc. in figure captions. When denoting sample sizes, it would be useful to know both the number of urine samples and the number of unique individuals that contributed to the dataset.

g. I would like to see more transparency about sample size, concerning the number of samples, from x individuals, in y study groups.

3. It would be great if authors can provide additional data that show possible differences in survival and/or behaviour between orphaned and non-orphaned immatures and further incorporate the reason for the maternal loss and the age at which mothers died into the analyses. An aged mother dying could have different effects compared to a prime-age mother dying. It is hugely surprising that behavioural data is completely excluded from the study after claiming to have followed individual animals for 6 or 12 hours per day. The manuscript would be quite a bit different from what we have reviewed here but tying the cortisol data to some concrete behavioural and survival observations would help contextualize the results. See specific comment from a reviewer below:

a. L294 briefly mentions survivorship bias. I would like to see a more thorough discussion of this. Did any individuals that were orphaned subsequently die? How were these handled? Are there enough to compare them to those that survived?

b. I wondered throughout the manuscript whether and how post-weaning survival could be included more directly to bring clarity to the role of cortisol/HPA regulation in fitness. I am not exactly sure what to suggest- but I think that directly discussing how differences in survival/reproduction may be related to HPA functioning in this population of chimpanzees, even if they are limited to qualitative comparisons, could improve the manuscript quite a lot.

[Editors' note: further revisions were suggested prior to acceptance, as described below.]

Thank you for resubmitting your work entitled "Early maternal loss leads to short- but not long-term effects on diurnal cortisol slopes in wild chimpanzees" for further consideration by *eLife*. Your revised article has been reviewed by 2 peer reviewers, and the evaluation has been overseen by a Reviewing Editor and George Perry as the Senior Editor.

The manuscript has been improved but there are some remaining issues that need to be addressed, as outlined below:

Your study examines the relationship between maternal loss and cortisol excretion among wild chimpanzees. It tests ideas from multiple fields to determine whether and how maternal loss is reflected in the alteration of the HPA axis. Three reviewers commented on your initial submission and we requested a revision of the manuscript.

You have clearly put a great deal of additional work into this revision and the manuscript is much improved from the prior submission. We find your responses to our previous concerns satisfying.

There are, however, a few areas that would still require work/clarification including some typos that should be fixed before we can make a decision on your submission. Note that this does not amount to a partial acceptance of your manuscript.

Introduction: We greatly appreciate the care that you have taken in integrating the suggestions of all reviewers and recrafting the introduction. This section does a wonderful job of setting up the study and the edits you made make for a very impactful series of arguments.

Methods/Results: There are still some spots that are difficult to follow in the results, and this section might require a bit more work than others. Consider the specific comments below:

L266-294: This section describing the models is still a bit confusing, especially the section about predictor variables and how they differ between models 1a/b, 2a/b. Later in the methods, the authors mention that they ran models for each sex separately, but that is not mentioned here.

I was slightly confused by this list of predictors at first thinking all predictors were used in each model. Is it possible to make it a little clearer that this is not the case? Perhaps something like "Each model contained one or more of the following test predictor variables" in line 283/284. In line 287 I think you mean model 1a rather than model 1b – this is probably the root of my confusion as it makes it seem like both years since maternal loss and orphan status as a categorical variable with three levels were included in the same model.

I don't have much specific advice to solve the problem, other than to say that it was difficult to follow each thread. Perhaps if each section was more simply dedicated to each model (like the longer methods section) rather than going back and forth between the things that were the same across models versus different? It's a lot to keep track of, so redundancy might be better for ease of interpretation in this case??

L315-16: I am not very familiar with Bayesian approaches and this section is unclear to me. In frequentist statistics effects sizes and variance explained are not the same things – could the authors clarify what they are reporting here and what it means?

L329-331: Did the authors directly test for categorical differences in morning cortisol or evening cortisol or are all of the comparisons here based on slope?

L362-365: The wording here "in particular in the early morning and the afternoon" is confusing given that the take-away is that cortisol had an upward slope and was, therefore, higher in the afternoon compared to the morning.

L374-389: The authors jump back and forth between describing life-history-based age categories (under 5 y.o. = infants, 5-8 = juveniles, 8-12 = adolescents) and referring to specific ages ("who lost their mother at 4 y.o."). That makes it difficult to parse whether and where they are using continuous versus categorical age predictions. It is especially difficult because the text describes things as one way or both ways, but the figures describe something firmly in the middle. Please revise these sections to make them clearer.

Discussion: L444-447: Higher morning and higher evening cortisol does not necessarily mean anything about slope (i.e. the am and pm increases could be equally leading to similar slopes, but higher average cortisol). I think it is important to specify exactly what the authors mean here- are orphans experiencing higher am, higher pm, and different slopes? If so how are the slopes different in layman's terms and which point is contributing to that difference in slope? It looks like the answer comes later (lines 457-58), but it still isn't so clear throughout the paragraph which parts of the results correspond to what theoretical models/predictions, and how. For instance, in L447-448: could the authors be more specific about how this finding aligns with the ACM?

L471-476: How often does food sharing happen with mothers and weaned offspring? It seems like the authors are asserting that calories from food sharing make up a significant portion of the juvenile chimpanzee diet. Is this the case? If so, that would seem different from other sites.

L532: One thing to be careful about in discussing adaptive calibration is that the model is more focused on the plasticity of the HPA axis than a change in the environmental conditions. In other words, a return to normal could reflect that the environment has adjusted-but the ACM predicts that the HPA readjusts itself during critical developmental/life history timepoints (e.g. adrenarche, puberty, pregnancy/parenthood) to account for environmental conditions. So that return to normal could be the HPA readjusting itself to essentially make what it previously considered a stressful environment led to less of a stress response kind of like making it a new normal.

Methods: Can authors add a bit more detail about the choices that they made in creating these models? This will be instructive for helping other scholars follow and match their methodology. For instance (L872-875), what is the difference between a regularizing prior and any other type of prior?

One general question: because I'm not so familiar with Bayesian LMM/GLMM, are there any guidelines or rules for limiting the number of predictor/control terms included in the models? The authors have clearly gone to great pains to control for things, so the concern would just be that including so many terms would exhaust the degrees of freedom for the number of individuals included in the study.

L772-774: If the models are fit separately for males and females, does that mean that 1a/b are four models? 1aMale, 1aFemale, 1bMale, 1bFemale? Were there any differences in results for males versus females?

L861-865: Is this a standard control for this field? It's unclear how including project as a random effect would account for things that aren't already controlled for using the other factors mentioned here: individual, year, individual_year, etc.

L888: Does this mean that all of the actual sample sizes were > 1000? Or something else? My understanding is that there were models, e.g. those with immatures only, that included fewer than 1000 samples?

---

## [Author Response]

Summary:This paper tests the biological embedding model by asking whether and how early maternal loss effects cortisol levels and diurnal cortisol slopes among wild chimpanzees at the Tai Forest, Cote d'Ivoire. The results suggest that maternal loss alters the HPA stress axis in wild chimpanzees, but these effects are not visible later in life. Authors suggest that the lack of a later life association between maternal loss and cortisol levels may be due to selective early mortality of individuals with high cortisol levels but did not provide any survival or behavioural data to show that orphans and non-orphans differ in any fitness-related traits other than cortisol.

We acknowledge that a robust analysis assessing the effect of modification in cortisol profiles on survival would be, as suggested, a strong addition to our study. Limited by only 9 confirmed deaths of immatures before maturity in our sample, we conducted a descriptive analysis.

In order to link the diurnal cortisol slopes and cortisol levels to fitness, we compared the average cortisol levels (random intercepts in Model 1a) and linear diurnal cortisol slopes (random linear slope for time of the day in Model 1a) of immatures who survived until maturity (i.e. until 12 years of age) to those of individuals who did not survive until maturity. We compared the diurnal cortisol slopes and cortisol levels of immatures who survived until maturity (i.e. until 12 years of age) (N = 23) and the ones who did not (N = 9). For 18 of the 50 immatures we do not have information on whether or not they reached maturity since they either did not reach maturity yet or may have emigrated to another group before maturity. The comparison revealed that the average intercept (cortisol levels) and linear term for time of day (i.e. cortisol diurnal linear slope) of immatures who survived until maturity, whether they were orphaned or not, were very similar (mean log cortisol levels ± SE = 3.686±0.005 vs. 3.687±0.008 and mean linear diurnal cortisol slope ± SE = -0.381±0.004 vs. -0.384±0.004). This lack of difference did not arise from a lack of variation in individual random intercepts or slopes which ranged from 3.539 to 3.761 and from -0.446 to -0.307 respectively. Superficially, this descriptive analysis above supports our conclusion that effect of maternal loss on circadian cortisol patterns in adult chimpanzees indicates a recovery over time rather than a sampling bias due to the death of orphans with altered cortisol profiles.

Nonetheless, we feel that this analysis is flawed for two reasons. First, due to the stochastic nature of death in the Tai population, sudden death such as from predation or anthrax, may not be associated with cortisol patterns in the preceding months, limiting any interpretation of these descriptive results. Second, given that it is usually not known when an individual will die, sampling within a few months of death was very rare. Thus, what we capture here is a comparison of longer term patterns (using samples across several years), which as we see from Model 1a and b, can change over time, especially in orphans. Also, the sample was too small to control for potentially influential factors such as dominance and orphan status. Hence, although we have tried to follow the reviewer’s advice here, unfortunately we do not have a robust enough dataset to include such an analysis in the paper, but hope to build a stronger dataset over the next years.

In addition, we also investigated descriptively the survival of orphans and non-orphans in our sample until maturity and found that the overall likelihood to survive until maturity for immature orphans was not substantially different from the one of non-orphans (69.2% for orphans versus 73.7% for non-orphans). At face value, additional information, albeit based on only 9 deaths, support our conclusion that the absence of effect of maternal loss in adult chimpanzees indicates a recovery over time rather than a sampling bias due to the death of orphans with altered cortisol profiles. However, we would not be comfortable to publish this result based on such a small sample (for survival analyses), especially as papers from two other chimpanzee populations show a different result, that orphan status does impact on survival (Nakamura et al. 2014; Stanton et al. 2020).

Furthermore, the association between cortisol and the HPA axis is in the opposite direction to that observed in humans and there seems to be no significant increase in cortisol in orphans compared to non-orphans.

We were also surprised that the direction of the effect on diurnal cortisol slopes for the orphans was opposite to that in most human studies. We are nevertheless confident in the meaningfulness of this effect since we could confirm this result using a Bayesian framework allowing us to investigate in more depth the level of uncertainty in the effect found (see details below).

Overall, the study is the result of extensive fieldwork and the number of samples collected is impressive. The subject is very interesting, and we generally agree that with an extensive reworking of the entire framework and analyses, it could be a good fit for eLife.

We thank the editor and the reviewer for this positive evaluation of our manuscript.

The analyses will benefit greatly if the authors use effect sizes and confidence intervals for inferences instead of p-values. This may solve the significance threshold issues. Moreover, the reliance on p-values seem to limit the value of the data. For example, authors suggest that results from model 1 should be treated with caution because the full model is not significantly different from the null model, but by relying on it as the key finding of the study without exploring effect sizes, it does not seem that they did exercise sufficient caution.

We agree that the reliance on p-value only does not allow full exploration of the degree of uncertainty or consistency in the effects investigated. Therefore, we reran the entirety of our analysis using a Bayesian framework that allows for more accurate estimations of uncertainty in effect estimates when using models with a complex random effect structure, such as in our study.

In the new analysis we also included, as suggested by the reviewers, a cubic term for time of the day and a categorical variable for orphan status with three levels (orphan for less than 2 years, orphan for more than 2 years, non-orphan). This new analysis highlighted that orphans who lost their mother less than two years ago (but not other orphans), had a consistently steeper diurnal cortisol slope (on average 58% steeper) than non-orphans. It also confirmed the effect of the age at maternal loss on immature diurnal cortisol slopes with immatures that lost their mother earlier in life having upward curving slopes, characterized by higher morning and afternoon levels. Finally, the new analysis using a cubic term for time of the day revealed that the diurnal cortisol slopes of immatures were affected by the time that passed since their mother died.

We report now 95% and 90% credible intervals for the estimates as well as the percentage of the posterior distribution which supports the direction of the estimate for each test predictor in all our models to add extra information for the reader on the degree of uncertainty of each effect. Following reviewers’ suggestions, we also report effect sizes whenever possible.

Please find more specific comments below:Essential revisions:1. Present results as effect sizes with confidence intervals and make inferences along the line of the percentage (or ng/ml) by which orphans differ from non-orphans and over time. This effect sizes can be more easily compared with results from human studies on cortisol. Please communicate findings more clearly and discuss exactly why the pattern in this Chimp population may be different from that in humans. Pay attention to the following comment from reviewers:

We thank the reviewers for this suggestion which improves the clarity of our manuscript. We have rewritten the entire result section based on the results of our new analyses (see above) and provide more information on the degree of uncertainty and magnitude of our results by systematically reporting now marginal R^2^ and conditional R^2^ for each model, the 95% and 90% credible intervals for each estimate, the percentage of the posterior distribution which confirms the effect of the estimate for each test predictor as well as, whenever possible, effect sizes. In particular, as mentioned above, we found that recently orphaned immature chimpanzees (i.e. individuals who lost their mother for less than 2 years) had diurnal cortisol slopes which were on average 58% steeper than non-orphans. We have clarified the discussion on why this effect differs in direction from most findings in humans.

a. Despite acknowledging that the "significance of these predictors should be interpreted with caution" because model 1a did not reach significance, the authors make very strong claims about the results in the discussion- and also feature the finding of that model in the title of the paper. That seems problematic to me- especially because the insignificant model results (more intense diurnal slopes among immature orphans) diverge from the expectations set forth by other works in humans and non-humans. The finding that this is to do with higher-than-expected morning cortisol is puzzling given that evening levels are generally considered more responsive or plastic. However, this could also be an artefact of fitting the models without the third-order term for time.2. The lack of significance could be due to insufficient sampling or a true lack of predictive power. Reviewers provide specific suggestions on how to reanalyse the data given the difficulty of collecting additional samples currently.a. Model 1B and figure 2 demonstrate that the cortisol response to maternal loss declines over time and that after 2 years it is no longer detectably different from non-orphans. The authors do not account for age since maternal loss in model 1A. If a considerable proportion of samples were from orphans that lost their mothers more than 2 years ago, this would reduce the likelihood of detecting a significant difference between orphans and non-orphans and potentially explain the lack of significance in the overall model. Crucially I think if model 1a was adjusted to separate out recent orphans from those that lost their mothers less recently this could enable the authors to better back up their claims at least in relation to changes in overall cortisol levels.

We are grateful to these constructive suggestions which improved the accuracy of our analysis. Following reviewers’ comments on how to modify the structure of model 1a (comments 1a and 2a) by incorporating a cubic term for time of the day and categorizing orphans between recently and non-recently orphans we have rerun a modified version of Model 1a using a Bayesian framework. This new analysis allowed us to report more precisely on the degree of uncertainty of the results (see above). We found that recently orphaned immature chimpanzees had consistently steeper diurnal cortisol slopes than non-orphan immatures. The difference in slope was however not consistent between orphans who had lost their mother for longer than 2 years and non-orphans. Therefore, it is possible, as the reviewer pointed out, that the lack of categorization of orphans into the two categories and of the non-inclusion of a cubic term for time of the day led to the lack of significance in our previous analysis.

b. The truth is that cortisol data are very messy and even though 300+ samples from 50 individuals might seem like a lot, it might turn out that it isn't enough to detect a signal. At other sites, cortisol levels and diurnal slopes shift with age- and this is true for humans as well. However, the slope should be more susceptible than more average levels so the authors might be able to make stronger conclusions based on average or time-corrected cortisol rather than focusing so much on a slope. Either way- though improved modelling to access slope or by setting slope aside and focusing on average cortisol- the data here certainly have a path to publication

We have modified our models to include a cubic term for time of the day and thus improve our modelling of the diurnal cortisol slopes. Using a Bayesian framework, we are now able to assess better the uncertainty in our results and report on it. We would like to point out that we have 846 samples in our first model (Model 1a) from 50 individuals not only 300.

In our analysis we set a threshold at three samples per year per individual since with only 2 samples an unusually high or low value would lead to dramatic changes in the slope steepness and direction. Mathematically, three sample is the bare minimum to calculate a meaningful slope and that is why we chose this threshold. However, it is important to note that the number of samples per individual per year is well above such a threshold, therefore we are confident that the slopes are estimated with accuracy. In fact, in model 1a we have on average 9 sample per year per individual. This information has been added to the manuscript (Lines 523-526).

c. My principal concerns with this paper, as written, revolve around the methods/results. First and foremost, I am not convinced that the authors have a sufficient sample size to evaluate the predictions/hypotheses outlined in the introduction. While 849 urine samples are a large number, and again, their efforts here should be commended, the sample spread is quite thin once it is spliced up into appropriate categories, especially considering how many samples were collected per individual year, on average. As the authors indicate throughout and especially when describing their modelling approach, cortisol is inherently a very noisy hormone impacted by a myriad of factors- including age in at least one other densely sampled chimpanzee community.

We agree that 849 samples may appear rather thin as a sample size. However, since, as mentioned above, the average number of sample collected per year per individual was 9, we are confident that our dataset allows for modelling meaningful slopes. Furthermore, we now revert to a Bayesian approach which provide more accurate estimates of the effect for models with limited sample size and complex random slope structures as ours.

Please note that, while reanalysing the data we realized that one individual in the year that it was orphaned, had less than three samples collected before and after the date that he was orphaned. Since the individual_year random slope structure in Model 1a is also separating the year when the individual is orphaned and the year when he is not (even within the same year) we had to exclude samples from that individual that year. Our final sample size still comprises 50 individuals but was slightly reduced to 846 samples.

I am also surprised that time of day was modelled quadratically. It is my understanding that humans, other populations of chimpanzees, and other mammals follow a sigmoidal curve which should be modelled with a third-order term as well. For these reasons, it is difficult to tell whether model 1A is not significant because of the insufficient sample or a true lack of predictive power. Additionally, I am concerned that the paper seems to focus so much on the results from a single model term in a model that did not reach significance.

We thank the reviewer for this suggestion allowing us to model our data more accurately. Following the reviewers’ suggestion, we have now incorporated a cubic term for time of the day in all our statistical models. That addition plus the new categorical variable for orphan status in model 1a separating orphans into recently and non-recently orphaned provide clear evidence for a robust and consistent effect of being recently orphaned on diurnal cortisol slopes: recently orphaned immatures had a diurnal cortisol slope on average 58% steeper than non-orphans. Since the 95% credible interval for this effect does not overlap 0 we can discuss this result with more certainty than the result of the previous model in the first version of the manuscript.

d. It would be useful to see some of the raw data- especially a plot showing cortisol values across the time of day. Regarding that third-order term- it isn't out of the question that T + T^2 would be sufficient, however, I do believe it's important to rule out that including T^3 does a better job.

We agree that it is important to show the raw cortisol data across the time of the day. We have now added a plot (Appendix 1-figure 1) showing the cortisol value for each sample in immature chimpanzees.

Please note that Figures 1, 2 and 3 show cortisol data for each hour of the day and also allow the reader to visualize raw cortisol variation throughout the day. In these Figures we decided to plot only 1 data point for each orphan category for each hour since it allows for better visualization of the daily variation and of the CI.

e. L449 As slopes are calculated for each individual each year, what is the mean number of samples per individual per year? 3 minimum seems very small for calculating a slope but if the average is considerably higher, then perhaps it is not an issue.

We are confident that our sample size is sufficient to evaluate individual diurnal cortisol slope yearly since the average number of samples collected for each individual each year is three times our cut-off criteria of three samples for the immatures (9 sample on average per individual per year and 50% of the ID_year comprise over 7 samples) and more than six times our cutoff criteria for the adult males (19 sample on average per individual per year and 50% of the levels for the ID_year variable comprise over 10 samples). This information has been added to the manuscript (Lines 523-526).

Please note that consistent inter-individual differences in diurnal cortisol slopes can be found using a smaller sample size than ours per individual per year (Sonnweber et al. 2018) so that we are confident that our sample size is large enough to provide meaningful results.

f. Please include more information about model results, sample size, etc. in figure captions. When denoting sample sizes, it would be useful to know both the number of urine samples and the number of unique individuals that contributed to the dataset.

As suggested by the reviewer we added information on model results and sample sizes in the figure caption for each figure.

g. I would like to see more transparency about sample size, concerning the number of samples, from x individuals, in y study groups.

We added information on the number of samples and of individuals for each model and for each orphan categories in the main text throughout the method and result sections. In addition, we added extra information on sample size to Table 1 (i.e. mean ± SE number of sample per individual for each orphan category for immature and mature males for each community).

3. It would be great if authors can provide additional data that show possible differences in survival and/or behaviour between orphaned and non-orphaned immatures and further incorporate the reason for the maternal loss and the age at which mothers died into the analyses. An aged mother dying could have different effects compared to a prime-age mother dying. It is hugely surprising that behavioural data is completely excluded from the study after claiming to have followed individual animals for 6 or 12 hours per day. The manuscript would be quite a bit different from what we have reviewed here but tying the cortisol data to some concrete behavioural and survival observations would help contextualize the results. See specific comment from a reviewer below:

Incorporating information on survival and maternal health and age is very interesting, and a future goal in our research. Unfortunately, our sample in this paper does not permit a meaningful analysis of this, as only 9 of the immatures died. As described above we have conducted a descriptive analysis of the cortisol slopes and cortisol levels of immatures surviving or not until maturity in our samples. However, as explained in details above, we do not feel confident to publish such an analysis since it is not robust enough.

As for the behavioural data, unfortunately most of the samples for immatures collected for this study were collected opportunistically alongside focal observation of mature individuals. Our research group only recently began collecting behavioural samples for immatures. Therefore, we cannot causally link behaviour to hormonal patterns in the context of this paper.

a. L294 briefly mentions survivorship bias. I would like to see a more thorough discussion of this. Did any individuals that were orphaned subsequently die? How were these handled? Are there enough to compare them to those that survived?

As mentioned above there was no obvious survival bias of non-orphan as compared to orphan immatures in this sample (although we say this cautiously as there was not a sufficient sample to run a survival analysis and other chimpanzee sites do show that orphans lose out on survival, Nakamura et al. 2014; Stanton et al. 2020), nor an obvious difference in diurnal cortisol slopes or average cortisol levels between immatures that survived until maturity and immatures that did not. As mentioned above we did not include this information into the manuscript since it may be misleading for the reader.

Orphans that died very soon after their mother died are not included in the sample since we were unable to collect at least 3 samples from them following maternal loss, and thus unable to determine their circadian cortisol pattern. That is probably why the youngest orphan in our sample was 4.1 years of age. We are aware that this may have constitute a bias and discuss this issue in the manuscript: specifically, we address that there may be a sensitive window for mammals during immaturity in which long-term permanent modification of the HPA axis functioning occurs (Lines 461-471).

b. I wondered throughout the manuscript whether and how post-weaning survival could be included more directly to bring clarity to the role of cortisol/HPA regulation in fitness. I am not exactly sure what to suggest- but I think that directly discussing how differences in survival/reproduction may be related to HPA functioning in this population of chimpanzees, even if they are limited to qualitative comparisons, could improve the manuscript quite a lot.

A recent publication of our working group on the same population of chimpanzees found that individuals that lost their mother before reaching maturity suffer some fitness cost in the form of lower reproductive success as an adult (Crockford et al. 2020). While our analysis suggests that loss of fitness in these orphans is probably not directly linked to HPA regulation, the finding of our study may provide some possible mechanism through which individuals are affected in their growth (Samuni et al. 2020) via short-term alteration to HPA axis functioning which may lead to later loss in reproductive success. As for survival, as explained above, there was no indication that orphans survived less than non-orphans nor that the individuals with altered diurnal cortisol slopes or higher cortisol levels where less likely to survive until maturity. Yet this is based on a descriptive investigation of the data and a proper analysis needs to be conducted with a larger sample.

References:

Crockford, C., Samuni, L., Vigilant, L. and Wittig, R. M. 2020. Postweaning maternal care increases male chimpanzee reproductive success. *Science Advances*, 6, eaaz5746.

Nakamura, M., Hayaki, H., Hosaka, K., Itoh, N. and Zamma, K. 2014. Brief communication: Orphaned male chimpanzees die young even after weaning. *American Journal of Physical Anthropology*, 153, 139–143.

Samuni, L., Tkaczynski, P., Deschner, T., Löhrrich, T., Wittig, R. M. and Crockford, C. 2020. Maternal effects on offspring growth indicate post-weaning juvenile dependence in chimpanzees (Pan troglodytes verus). *Frontiers in Zoology*, 17, 1.

Sonnweber, R., Araya-Ajoy, Y. G., Behringer, V., Deschner, T., Tkaczynski, P., Fedurek, P., Preis, A., Samuni, L., Zommers, Z., Gomes, C., Zuberbühler, K., Wittig, R. M. and Crockford, C. 2018. Circadian rhythms of urinary cortisol levels vary between individuals in wild male chimpanzees: A reaction norm approach. *Frontiers in Ecology and Evolution*, 6,

Stanton, M. A., Lonsdorf, E. V., Murray, C. M. and Pusey, A. E. 2020. Consequences of maternal loss before and after weaning in male and female wild chimpanzees. *Behavioral Ecology and Sociobiology*, 74, 22.

[Editors' note: further revisions were suggested prior to acceptance, as described below.]

[…] There are, however, a few areas that would still require work/clarification including some typos that should be fixed before we can make a decision on your submission. Note that this does not amount to a partial acceptance of your manuscript.Introduction: We greatly appreciate the care that you have taken in integrating the suggestions of all reviewers and recrafting the introduction. This section does a wonderful job of setting up the study and the edits you made make for a very impactful series of arguments.

Thank you very much for the positive evaluation of our introduction. We are happy to hear that you appreciate the way we incorporated your previous comments. Thank you again for giving us the tools to strengthen the manuscript.

We have reworked the unclear parts of the manuscript, redrawn all the figures and corrected the typos to improve clarity and readability, and explain this in detail below.

Methods/Results: There are still some spots that are difficult to follow in the results, and this section might require a bit more work than others. Consider the specific comments below:L266-294: This section describing the models is still a bit confusing, especially the section about predictor variables and how they differ between models 1a/b, 2a/b. Later in the methods, the authors mention that they ran models for each sex separately, but that is not mentioned here.

We apologize for the confusion. We agree that the phrasing was misleading. All our immature models include samples from both males and females, whereas only males were included in the mature models. To clarify the statistical analyses further, we have changed the model names to provide an intuitive description of each model. Model 1a and 1b have been replaced by “all immature” and “immature orphan”, and models 2a and 2b have been replaced by “all adult male” and “adult male orphan” models.

We have edited the text accordingly: “We used a series of Bayesian Linear Mixed Models (LMMs) to test our predictions regarding the effect of maternal loss on overall cortisol levels and diurnal slopes (jointly constituting the cortisol profile). We first tested these effects in socially immatures (i.e. males and females <12 years of age because prior to 12 years, chimpanzees associate primarily with their mother, Reddy and Sandel 2020). Secondly, we tested these effects in mature males (i.e. males >= 12 years of age)” (Line 238-242).

We have also modified the related paragraph in the result section (Lines 821-828).

I was slightly confused by this list of predictors at first thinking all predictors were used in each model. Is it possible to make it a little clearer that this is not the case? Perhaps something like "Each model contained one or more of the following test predictor variables" in line 283/284. In line 287 I think you mean model 1a rather than model 1b – this is probably the root of my confusion as it makes it seem like both years since maternal loss and orphan status as a categorical variable with three levels were included in the same model.

We have rephrased this section to make the presentation of our models and of the different predictor variables clearer. We present now the set of predictor variables for each model separately (Lines 273-285).

I don't have much specific advice to solve the problem, other than to say that it was difficult to follow each thread. Perhaps if each section was more simply dedicated to each model (like the longer methods section) rather than going back and forth between the things that were the same across models versus different? It's a lot to keep track of, so redundancy might be better for ease of interpretation in this case??

We modified our presentation of the model parameters to clarify which variables are included in each model (Lines 273-285). As mentioned above we also changed the name of models to improve clarity (i.e. we replaced Model 1a, 1b, 2a, 2b by “all immature”, “immature orphan”, “all adult male” and “adult male orphan” models).

L315-16: I am not very familiar with Bayesian approaches and this section is unclear to me. In frequentist statistics effects sizes and variance explained are not the same things – could the authors clarify what they are reporting here and what it means?

We report the percentage of the variance explained by the random and fixed effect. You are correct that it is not the same thing as the effect size. Effect sizes are reported elsewhere in the text (e.g. “On average, recently orphaned individuals had a diurnal cortisol slope 58% steeper than non-orphans” Lines 366-367). We have removed mention of effect size in this sentence and simply mention that we report the percentage of variance explained by the fixed and random effects.

L329-331: Did the authors directly test for categorical differences in morning cortisol or evening cortisol or are all of the comparisons here based on slope?

All the comparisons are based on the meaningful differences in slopes and on visual assessment based on the figures depicting the model predictions. We have clarified that in the text (Lines 345-349).

L362-365: The wording here "in particular in the early morning and the afternoon" is confusing given that the take-away is that cortisol had an upward slope and was, therefore, higher in the afternoon compared to the morning.

We have modified our phrasing to clarify our point: “A visual inspection of the data and the model prediction lines reveals that cortisol levels of immature orphans who lost their mother recently had higher cortisol levels throughout the day (orange squares in Figure 2) as compared to immature orphans who lost their mother several years ago (green triangles in Figure 2). The difference in cortisol levels between recently and non-recently orphaned individuals was most evident during early morning and late afternoon (Figure 2).“ (Lines 396-402).

L374-389: The authors jump back and forth between describing life-history-based age categories (under 5 y.o. = infants, 5-8 = juveniles, 8-12 = adolescents) and referring to specific ages ("who lost their mother at 4 y.o."). That makes it difficult to parse whether and where they are using continuous versus categorical age predictions. It is especially difficult because the text describes things as one way or both ways, but the figures describe something firmly in the middle. Please revise these sections to make them clearer.

We modified all the figures to include age categorical intervals that correspond to the life history stages mentioned in the text and are in line with the main text.

Discussion: L444-447: Higher morning and higher evening cortisol does not necessarily mean anything about slope (i.e. the am and pm increases could be equally leading to similar slopes, but higher average cortisol). I think it is important to specify exactly what the authors mean here- are orphans experiencing higher am, higher pm, and different slopes? If so how are the slopes different in layman's terms and which point is contributing to that difference in slope? It looks like the answer comes later (lines 457-58), but it still isn't so clear throughout the paragraph which parts of the results correspond to what theoretical models/predictions, and how. For instance, in L447-448: could the authors be more specific about how this finding aligns with the ACM?

We have edited most of the paragraph to be more specific as to how our results correspond with the predictions of the ACM, and to improve the description of the results found across orphan categories. We specify that the higher early or afternoon cortisol levels for immatures orphaned early as compared to other orphans are derived from visual inspection of the model lines:

“Orphans experiencing maternal loss at younger ages had a diurnal cortisol slope differing from the immatures orphaned when older, in particular in the quadratic term for time of the day (i.e. in how the slope curved). […] In fact, diurnal cortisol slopes are indicative of the general functioning of the HPA axis (Karlamangla et al. 2019) and our results indicate that the diurnal cortisol slope of immature chimpanzees undergo different levels of changes depending on the age at which they experience maternal loss, with more substantial deviations from mother-raised offspring pattern in immatures orphaned earlier in life.” (Lines 501-526).

L471-476: How often does food sharing happen with mothers and weaned offspring? It seems like the authors are asserting that calories from food sharing make up a significant portion of the juvenile chimpanzee diet. Is this the case? If so, that would seem different from other sites.

We have modified the sentence to clarify that the food sharing between the mother and weaned offspring is occasional (Line 557).

L532: One thing to be careful about in discussing adaptive calibration is that the model is more focused on the plasticity of the HPA axis than a change in the environmental conditions. In other words, a return to normal could reflect that the environment has adjusted-but the ACM predicts that the HPA readjusts itself during critical developmental/life history timepoints (e.g. adrenarche, puberty, pregnancy/parenthood) to account for environmental conditions. So that return to normal could be the HPA readjusting itself to essentially make what it previously considered a stressful environment led to less of a stress response kind of like making it a new normal.

We have modified our argumentation relating our results to the ACM model in the mentioned paragraph and in other sections of the discussion. We have rephrased the specific paragraph as follows:

“The re-establishment of a normal functioning of the HPA axis in mature male chimpanzees but also in immatures orphaned for more than two years may reflect a form of recovery in those individuals. […] The lack of apparent long-term effects of maternal loss on immature chimpanzee physiology could thus be indicative of ameliorations in the environment of these orphans in the years following maternal loss, possibly in terms of improved access to social support and food.” (Lines 617-626).

Methods: Can authors add a bit more detail about the choices that they made in creating these models? This will be instructive for helping other scholars follow and match their methodology. For instance (L872-875), what is the difference between a regularizing prior and any other type of prior?

We added information regarding our choice of priors to the method section:

“We chose weakly regularising prior for the fixed effects since they give less weight to outlier data points and therefore help constrain model predictions to biologically meaningful estimates and CI (Lemoine 2019).” (Lines 946-948).

One general question: because I'm not so familiar with Bayesian LMM/GLMM, are there any guidelines or rules for limiting the number of predictor/control terms included in the models? The authors have clearly gone to great pains to control for things, so the concern would just be that including so many terms would exhaust the degrees of freedom for the number of individuals included in the study.

The general guideline is to have at least 10 data points per degree of freedom in the model (i.e. per levels of predictor variable). Our model with the smallest sample size and the largest number of predictor variables is Model 1b (now called “immature orphan model”) in which we use 393 data points for 18 predictors (including the interaction terms) resulting in over 20 data points per predictor, well above the recommended threshold. We therefore do not think that model complexity is a concern for our analyses.

L772-774: If the models are fit separately for males and females, does that mean that 1a/b are four models? 1aMale, 1aFemale, 1bMale, 1bFemale? Were there any differences in results for males versus females?

Apologies for the confusion. We conducted two sets of analyses, either including samples from immature individuals (previously named model 1a and 1b) or mature individuals (previously named model 2a and 2b). The set of analyses of immatures included samples from both males and females in the same analysis, whereas the set of analyses of mature individuals included only males. We have clarified this point in the text:

“We used a series of Bayesian Linear Mixed Models (LMMs) to test our predictions regarding the effect of maternal loss on overall cortisol levels and diurnal slopes (jointly constituting the cortisol profile). […] Secondly, we tested these effects in mature males in the *all adult male* and *adult male orphan model*s (i.e. four models in total).” (Lines 821-825).

L861-865: Is this a standard control for this field? It's unclear how including project as a random effect would account for things that aren't already controlled for using the other factors mentioned here: individual, year, individual_year, etc.

In this study, we used a longitudinal dataset spanning 20 years, meaning that our data reflects the combined research effort of several observers, each having different research questions (e.g. some observers targeted urine sample collection after aggression while others collected samples opportunistically). Therefore, the sample collection is not fully random but may be biased by the question under investigation. To account for this variation, we have added the “project” as a random effect in our analyses. We have clarified this aspect in the manuscript:

“Finally, our hormonal dataset included samples collected by different observers with different research interests (hereafter project). […] Thus, to account for potential variation in cortisol levels that may be a result of inter-observer project bias, we added the ‘project’ type as an additional random factor.” (Lines 930-936).

L888: Does this mean that all of the actual sample sizes were > 1000? Or something else? My understanding is that there were models, e.g. those with immatures only, that included fewer than 1000 samples?

In Bayesian analysis, the effective sample size reflects the number of independent samples that were drawn from the posterior distribution to calculate our estimates using MCMC processes. Effective sample size here reflects the amount of autocorrelation within the chains. It is not related to sample size (i.e. it is not related to the number of actual data points that we used in our models). We have specified the later point in the manuscript:

“Please note that the effective sample size is a measure of autocorrelation and does not correspond to the number of data points that were used for each model (namely 393, 846, 2184, and 769 for the all immature, the orphan immature, the all adult male and the adult male orphan models respectively)” (Lines 985-988).